# El Niño Southern Oscillation influence on the Asian summer monsoon anticyclone

Xiaolu Yan[1,2], Paul Konopka[1], Felix Ploeger[1], Mengchu Tao[1], Rolf Müller[1], Michelle L. Santee[3], Jianchun Bian[2,4], and Martin Riese[1]

[1]Forschungszentrum Jülich (IEK-7: Stratosphere), Jülich, Germany
[2]Key Laboratory of Middle Atmosphere and Global Environment Observation (LAGEO), Institute of Atmospheric Physics, Chinese Academy of Sciences, Beijing, China
[3]Jet Propulsion Laboratory, California Institute of Technology, Pasadena, CA, USA
[4]College of Earth Science, University of Chinese Academy of Sciences, Beijing, China

*Correspondence to:* Paul Konopka (p.konopka@fz-juelich.de) and Jianchun Bian (bjc@mail.iap.ac.cn)

**Abstract.** We analyze the influence of the El Niño Southern Oscillation (ENSO) on the atmospheric circulation and the mean ozone distribution in the tropical and sub-tropical UTLS region. In particular, we focus on the impact of ENSO on the on-set of the Asian summer monsoon (ASM) anticyclone. Using the Multivariate ENSO Index (MEI), we define climatologies (composites) of atmospheric circulation and composition in the months following El Niño and La Niña (boreal) winters and investigate how ENSO-related flow anomalies propagate into spring and summer. To quantify differences in the divergent and non-divergent part of the flow, the velocity potential (VP) and the stream function (SF), respectively, are calculated from the ERA-Interim reanalysis in the vicinity of the tropical tropopause at potential temperature level $\theta$=380 K. While VP quantifies the well-known ENSO anomalies of the Walker circulation, SF can be used to study the impact of ENSO on the formation of the ASM anticyclone, which turns out to be slightly weaker after El Niño than after La Niña winters. In addition, stratospheric intrusions around the eastern flank of the anticyclone into the Tropical Tropopause Layer (TTL) are weaker in the months after strong El Niño events due to more zonally symmetric subtropical jets than after La Niña winters. By using satellite (MLS) and in-situ (SHADOZ) observations and model simulations (CLaMS) of ozone, we discuss ENSO-induced differences around the tropical tropopause. Ozone composites show more zonally symmetric features with less in-mixed ozone from the stratosphere into the TTL during and after strong El Niño events and even during the formation of the ASM anticyclone. Such isentropic anomalies are overlaid with the well-known anomalies of the faster/slower Hadley and Brewer Dobson circulation after El Niño/La Niña winter, respectively. The duration and intensity of El Niño related anomalies may be reinforced through the late summer and fall if the El Niño conditions last until the following winter.

## 1  Introduction

El Niño and La Niña are opposite phases of the El Niño Southern Oscillation (ENSO) which originate from the coupled interaction between the tropical Pacific and the overlying atmosphere (e.g., Bjerknes, 1929; Wang and Picaut, 2004; Roxy et al., 2015). ENSO is widely recognized as a dominant mode of the Earth's climate variability (McPhaden et al., 2006). In the

troposphere, ENSO manifests in the anomalies of the zonal distribution of convection which are triggered by a positive (El Niño) and negative (La Niña) sea surface temperature (SST) anomaly in the central and eastern Pacific (Philander et al., 1989). The SST anomalies typically peak during the Northern Hemisphere (NH) winter time (hereafter, seasons refer to the NH), but prolonged events may last for months or years (Moron and Gouirand, 2003; McPhaden, 2015).

Strong El Niño events disrupt the Walker circulation and lead to its breakdown during the warm ocean phases (Wang et al., 2002). Strong El Niño events also propagate upwards above the tropopause by accelerating the Brewer Dobson (BD) circulation and moistening the stratosphere (Scaife et al., 2003; Randel et al., 2009; Calvo et al., 2010). Using satellite observations and model simulations of water vapour and mean age of air, Konopka et al. (2016) have recently shown that wet (dry) and young (old) tape-recorder anomalies propagate upwards in the tropical lower stratosphere in the months following El Niño (La
Niña). They found that these anomalies are around $+0.3$ $(-0.2)$ ppmv and $-4$ $(+4)$ months for water vapour and age of air, respectively.

The Asian summer monsoon (ASM) anticyclone is a dominant feature of the circulation in the upper troposphere lower stratosphere (UTLS) during summer (Dethof et al., 1999; Randel and Park, 2006; Park et al., 2007). This nearly stationary anticyclone extends well into the lower stratosphere up to about 18 km (or $\theta = 420$ K) and effectively isolates the air masses of tro-
pospheric origin inside from the much older, mainly stratospheric air outside this anticyclone (Park et al., 2008; Ploeger et al., 2015). This anticyclone has been repeatedly identified as a key pathway for stratosphere-troposphere exchange (STE) in summer and fall, both quasi-isentropically into the lowermost stratosphere and into the upper branch of the BD circulation, especially for water vapour and pollutants entering the global stratosphere (Bannister et al., 2004; Fueglistaler et al., 2005; Fu et al., 2006; Randel et al., 2010; Wright et al., 2011; Vogel et al., 2016; Ploeger et al., 2017).

Generally, enhanced isentropic STE between the extratropics and tropics is caused by the monsoon systems, in particular by the ASM during NH summer (Dunkerton, 1995; Chen, 1995). Haynes and Shuckburgh (2000) showed that, indeed, the subtropical jet acting as a transport barrier between the extratropics and tropics weakens during NH summer. Consequently, enhanced isentropic transport occurs in both directions, out of the tropics and from the extratropics into the tropics (termed in-mixing, in the following). Related stratospheric signatures can be found in the Tropical Tropopause Layer (TTL) as diag-
nosed from NASA Aura Microwave Limb Sounder (MLS) observations of HCl and ozone (Santee et al., 2011, 2017). This in-mixed ozone contributes to more than half of the annual cycle of ozone in the upper part of the TTL (Konopka et al., 2010; Ploeger et al., 2012). Enhanced quasi-isentropic transport from the tropics to the midlatitude lowermost stratosphere driven by the ASM is also clearly observed both for tracers and water vapour (Ploeger et al., 2013; Müller et al., 2016; Vogel et al., 2016; Rolf et al., 2018)

A regionally-resolved view on the processes coupling ENSO with the stratosphere, mainly during the winter and spring, has been adopted in several previous studies (Krüger et al., 2008; Liess and Geller, 2012; Garfinkel et al., 2013; Konopka et al., 2016). However, there are only a few publications investigating the impact of ENSO on the ASM anticyclone and on the related STE (Ju and Slingo, 1995; Kawamura, 1998; Wang et al., 2013). This is in contrast with a large number of investigations connecting ENSO with the tropospheric variability of the ASM, such as weather patterns and precipitation, which have a long
tradition starting with the pioneering studies of Walker (1923) and Bjerknes (1969).

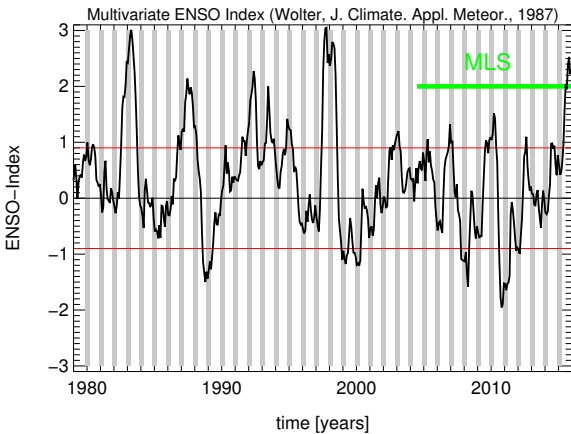

**Figure 1.** Multivariate ENSO Index (MEI) from the NOAA Climate Diagnostic Center, http://www.esrl.noaa.gov/psd/enso/mei (Wolter, 1987). The red lines denote the threshold values (±0.9) defining the El Niño (positive) and La Niña (negative) composites as used in this paper. Grey shading shows winter seasons (December-February, DJF). As in Konopka et al. (2016).

In this study, we investigate how the ENSO winter signal propagates into the following seasons. In particular, we characterize the impact of ENSO on the upper branches of the Walker and Hadley circulation in the UTLS. We focus on the ASM anticyclone, its strength as well as its efficiency for in-mixing of stratospheric ozone into the TTL. We investigate how long through the year ENSO related differences can last in the TTL both in the meteorological reanalysis as well as in long-term satellite, Lagrangian model and in-situ ozone data. Section 2 discusses data and methods for our analysis. Section 3 describes the seasonal propagation of ENSO anomalies. Section 4 quantifies the influence of ENSO anomalies on the seasonality of ozone in the TTL. Section 5 provides the discussion. The last section gives the conclusions.

## 2 Data and methods

There are several indices to indicate the phase of ENSO, and they are highly correlated (Pumphrey et al., 2017). Here, the Multivariate ENSO Index (MEI, Fig. 1) from the NOAA (National Oceanic and Atmospheric Administration) Climate Diagnostic Center, http://www.esrl.noaa.gov/psd/enso/mei, is used to quantify the ENSO variability (Wolter and Timlin, 2011). MEI is calculated based on sea surface pressure, zonal and meridional components of the surface wind, SST and total cloudiness fraction of the sky over the tropical Pacific. The two phases of ENSO typically show pronounced features in late fall, winter and early spring (Moron and Gouirand, 2003; McPhaden, 2015). Correspondingly, MEI shows peak values during this period. Negative and positive values of MEI quantify La Niña and El Niño events, respectively.

Hereafter, we define two winter composites (December-February, DJF) of ENSO events by the condition MEI$< -0.9$ for La Niña and MEI$>0.9$ for El Niño (red lines in Fig. 1) as discussed in Konopka et al. (2016). The winter months defining these two composites (17 months for 6 La Niña events and 28 months for 12 El Niño) are listed in Table 1. The quasi-biennial os-

| La Niña | | | El Niño | | |
|---|---|---|---|---|---|
| Year | Months | QBO | Year | Months | QBO |
| 1988/1989 | DJF | W | 1979/1980 | D | E |
| 1998/1999 | DJF | E | 1982/1983 | DJF | W |
| 1999/2000 | DJF | W | 1986/1987 | DJF | E |
| 2007/2008 | DJF | E | 1987/1988 | DJ | W |
| 2010/2011 | DJF | W | 1991/1992 | DJF | E |
| 2011/2012 | DJ | W/E | 1992/1993 | F | W |
| | | | 1994/1995 | DJF | E/W |
| | | | 1997/1998 | DJF | W |
| | | | 2002/2003 | DJF | W |
| | | | 2006/2007 | DJ | W |
| | | | 2009/2010 | DJF | E |
| | | | 2015/2016 | D | W |

**Table 1.** List of all relevant La Niña and El Niño winter months during the period of 1979-2015. In total there are 17 and 28 months for the La Niña and El Niño composites, respectively, which are listed above (D/J/F for December, January and February). Within the La Niña composite there are 7 months in the easterly phase (E) and 10 months in the westerly phase (W) of the QBO (defined by 30 day smoothed equatorial wind at 50 hPa). For El Niño composites 11 months are in the easterly phase and 17 months in the westerly phase. The underlined years mark the long-lasting El Niño episodes (for details, see text). As in Konopka et al. (2016).

cillation (QBO) phase during the considered months is also listed (http://www.cpc.ncep.noaa.gov/data/indices/qbo.u50.index) and shows that our composites are only weakly biased by the westerly phase.

El Niño episodes which last over the whole following year, are selected as the special long-lasting El Niño cases and underlined in Table 1 with black color (like during 1987 and 1992). The exceptional El Niño in 1982, which starts in spring 1982 and lasts until fall 1983, is also considered as a long-lasting El Niño case (underlined with dashed black in Table 1). These 3 cases as well as the influence of QBO on the results will be separately discussed in section 5.

To study the effect of strong ENSO winters on the UTLS in the following months, we also consider climatologies of "shifted" composites for different seasons (DJF, JFM, FMA, MAM, AMJ, MJJ, and JJA), e.g., AMJ represents 4 months after ENSO winters (DJF). The mean value of a composite is defined from the averaged monthly means of its elements. Monte Carlo significance test is used to investigate whether the La Niña and El Niño composites are statistically different or not. Monte Carlo significance test procedures consist of the comparison of the observed data with random samples generated in accordance with the hypothesis being tested (Hope, 1968). We call two (La Niña and El Niño) composites statistically different when the significance Monte Carlo test for their difference is passed at a 95% confidence level after at least 1000 iteration steps.

To quantify ENSO anomalies in the climatological flow patterns, stream function (SF) $\psi$ and velocity potential (VP) $\chi$ are calculated (Tanaka et al., 2004) using meteorological data from ERA-Interim reanalysis during 1979-2015 (Dee et al.,

2011). According to the Helmholtz theorem, an arbitrary 2D horizontal flow $\mathbf{u} = (u, v)$ can be separated into a non-divergent (i.e. rotational) part $\mathbf{u_a}$ with $\nabla \cdot \mathbf{u_a} = 0$ and a divergent (i.e. irrotational) part $\mathbf{u_b}$ with $\nabla \times \mathbf{u_b} = 0$, i.e.,

$$\mathbf{u} = \mathbf{u_a} + \mathbf{u_b} = \mathbf{k} \times \nabla \psi + \nabla \chi, \tag{1}$$

where both parts can also be expressed in terms of the potentials $\psi$ and $\chi$. Here, $\mathbf{k}$ denotes the unit vector perpendicular
to the considered 2D surface. SF and VP are scalar quantities which are easy to plot and widely applied in meteorology and oceanography to represent large scale flow fields (see e.g., Evans and Allan, 1992; Kunze et al., 2016). SF quantifies the position and strengths of the cyclones and anticyclones. Following Tanaka et al. (2004), we use VP to represent the Walker circulation and the zonal mean of VP to quantify the Hadley circulation. SF and VP will be divided into El Niño and La Niña composites as described above.

Ozone distributions are used to validate our diagnostic of the flow and to understand the effect of ENSO on the atmospheric composition in the UTLS region. MLS ozone data (version 4.2) and the Hilo (Hawaii) ozonesonde data from Southern Hemisphere ADditional OZonesondes (SHADOZ, Thompson et al. (2007)) are used (see http://croc.gsfc.nasa.gov/shadoz) as references. MLS measurements provide 8/6 months of data for the 3/3 La Niña/El Niño episodes from 2004 to 2015. Respectively there are 14/11 months of data for the 5/5 La Niña/El Niño events from SHADOZ ozondesondes covering the period
1998-2015. Chemical Lagrangian Model of the Stratosphere (CLaMS) simulations (McKenna et al., 2002; Konopka et al., 2004; Pommrich et al., 2014) driven by the ERA-Interim reanalysis are used to obtain robust statistical composites of ozone (with the same number of La Niña/El Niño months as for SF and VP). Outgoing long-wave radiation (OLR) monthly data from NOAA during 1979-2015 complete our analysis as a proxy for deep convection
(see https://www.esrl.noaa.gov/psd/data/gridded/data.interp_OLR.html).

## 3 ENSO anomalies at the tropical tropopause from winter to summer

In this section, we use the composites of the stream function (SF) and the velocity potential (VP) introduced above to illustrate some ENSO-related differences in the mean flow properties around the tropical tropopause.

### 3.1 Cyclones and anticyclones

Seasonal variations of SF after strong La Niña and El Niño winters are shown in Fig. 2. Here, respective climatologies are
plotted at the potential temperature level $\theta$=380 K, which roughly marks the tropopause in the tropics and in the extratropics separates the overworld from the lowermost stratosphere (Holton et al., 1995; Gettelman et al., 2011). The panels in Fig. 2 start from the winter (top, DJF) and end with the summer distribution (bottom, JJA).

Because the divergent part of the flow at $\theta$=380 K is very small compared to its rotational part, isolines of SF approximate the climatological streamlines whereas strongest horizontal gradients of SF describe the highest flow velocities. The anticyclones
are represented by positive and negative SF values in NH and Southern Hemisphere (SH), with highest and lowest values corresponding to their centers, respectively. During DJF, the flow in the tropical UTLS between 60° E and 120° W is dominated

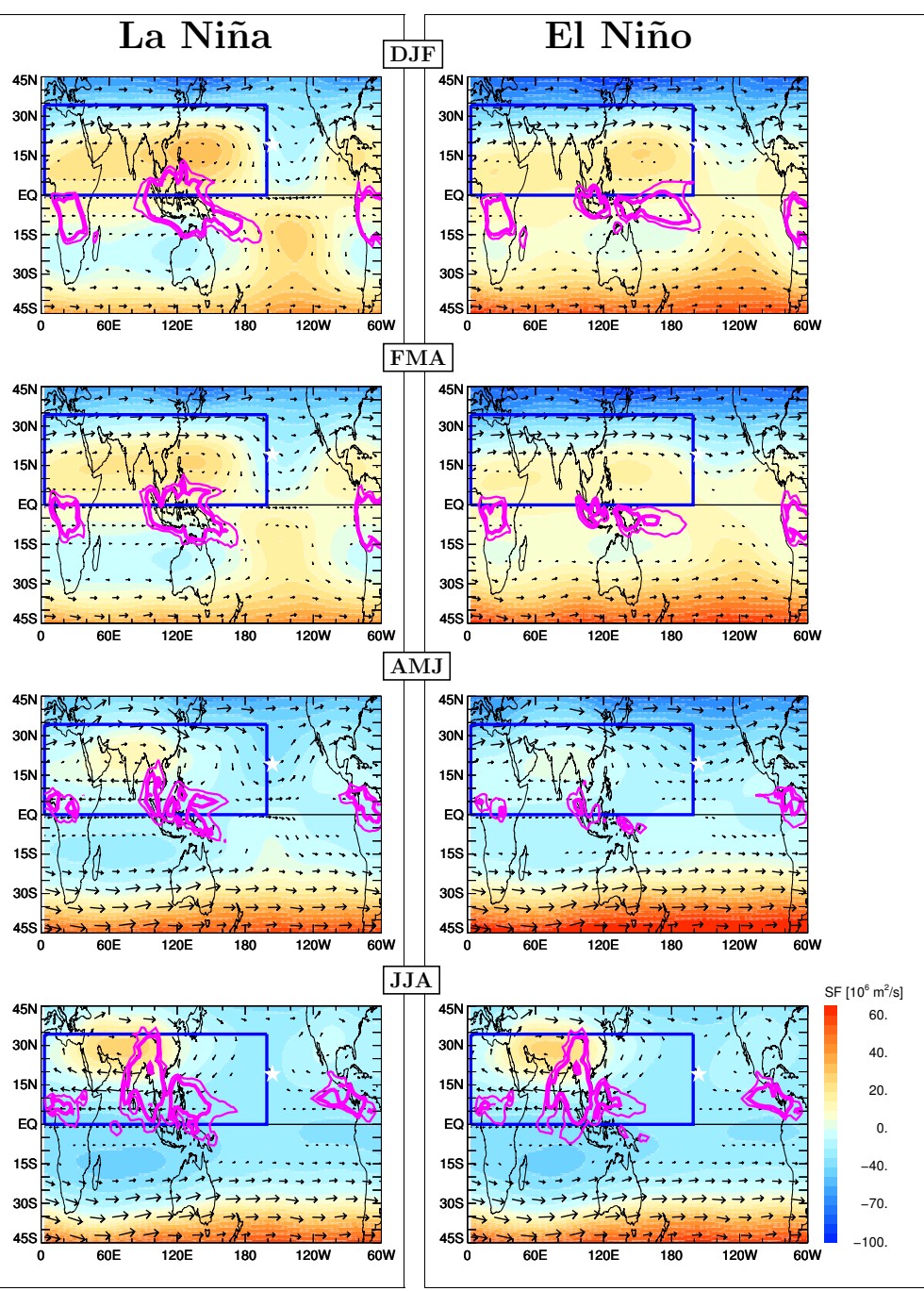

**Figure 2.** Climatologies (composites) of the stream function (SF, in $10^6$ m$^2$/s) at $\theta$=380 K calculated from ERA-Interim (1979-2015) for months following La Niña (left) and El Niño (right) winters until summer (from top to bottom). The arrows represent the rotational horizontal wind. Magenta isolines indicate the strong convection regions based on OLR (thick and thin lines represent 210 and 220 W/m$^2$ contours, respectively). The blue rectangles mark the locations of strong anticyclone in NH (for details, see text). Hereafter, the star in the figure marks the location of the SHADOZ station (Hilo, Hawaii) where long-term ozonesonde observations are available (see section 4.3).

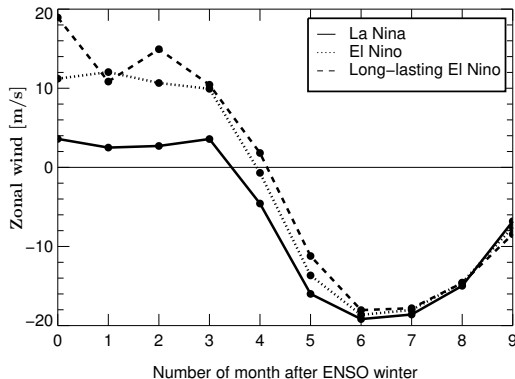

**Figure 3.** Mean zonal wind in the tropics over south-east Asia ([5° N, 20° N; 40° E, 120° E]) following La Niña, El Niño, and long-lasting El Niño winters at 200 hPa. The transition from positive to negative values marks the onset of the Asian summer monsoon (ASM). The zero mark on the x-axis denotes the middle of the DJF season (i.e., 15 January).

by two equatorially symmetric anticyclones resembling the well-known (symmetric) Matsuno-Gill solution with the heat source from convection located symmetrically over the equator (Matsuno, 1966; Gill, 1980; Highwood and Hoskins, 1998).

The climatological sources of heat can be approximated by the lowest values of the OLR. The analogous composites for OLR (magenta contours in Fig. 2) as for the SF are built with respect to La Niña and El Niño conditions. Thus, following the symmetric Matsuno-Gill solution as a proxy, the relevant latent heat sources for the anticyclones originate mainly in the western Pacific, especially during La Niña, and these sources are partially shifted to the east during El Niño events.

Over the course of the following 6 months, as the inter-tropical convergence zone (ITCZ) moves northwards, these two anti-cyclones shift to the north-west roughly following the position of convection (Highwood and Hoskins, 1998). The anticyclone in the NH intensifies, starting in May and June, and forms the well-known Asian summer monsoon (ASM) anticyclone during NH summer. In addition a weaker anticyclone in the SH can also be diagnosed. Thus, the summer configuration resembles more a superposition of a symmetric and anti-symmetric Matsuno-Gill solution (Gill, 1980; Zhang and Krishnamurti, 2006).

Now we discuss the differences in the large-scale flow in the UTLS caused by ENSO (i.e. differences between the left and the right column of Fig. 2). The most striking difference in DJF is a much weaker meridional disruption of the subtropical jets during El Niño than during La Niña winters, mainly in the NH subtropics between 170° E and 70° W. At the lower levels (not shown), such stratospheric intrusions coincide with regions of the so called "westerly ducts", which are much weaker during El Niño (Waugh and Polvani, 2000).

Furthermore, the equatorially symmetric anticyclones are more pronounced for the La Niña composites due to stronger and more localized convection in the western Pacific. These differences are also present during FMA, become smaller during AMJ and disappear during JJA mainly because forcing of the summer dynamics, especially of the ASM, is only weakly related to the winter forcing.

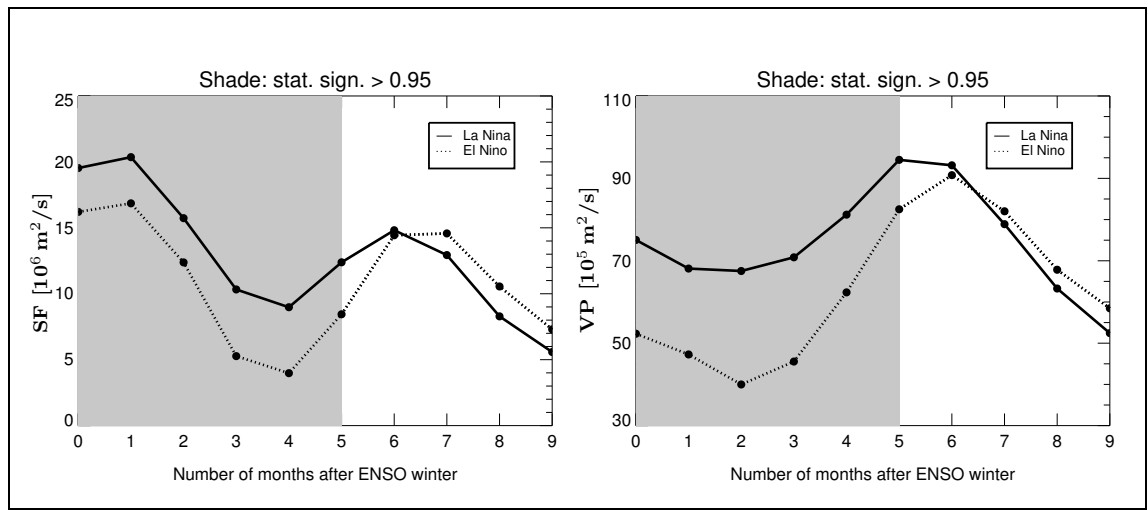

**Figure 4.** The average value of the stream function (left) in the domain of [0° N, 35° N; 0° E, 160° W] and velocity potential (right) in the domain of [30° S, 40° N; 90° E, 140° W] for La Niña (solid line) and El Niño (dotted line) composites at $\theta$=380 K. The grey shading region denotes the period with statistically significant differences between the two composites.

The mean climatological anticyclone in AMJ (Fig. 2) is at the very beginning phase of ASM anticyclone after El Niño winters, while it develops quickly and becomes stronger after La Niña winters. We use the transition of the upper-level (at 200 hPa) flow from westerly to easterly over south-east Asia [5° N, 20° N; 40° E, 120° E] to characterize the onset of the monsoon as discussed in Ju and Slingo (1995). By using the shifted composites as in the previous section, it turns out that the
5   onset of the ASM after La Niña is about a half month earlier than after El Niño (Fig. 3). The difference in SF between La Niña and El Niño composites lasts from winter (DJF) to early summer (AMJ) and becomes insignificant in summer (JJA) as noted earlier.

To prove the statistical significance of the ENSO anomalies in the SF composites, we compare their mean values averaged over a representative region shown as a blue rectangle in Fig. 2. The domain, defined as [0° N, 35° N; 0° E, 160° W], contains
10   the NH anticyclone from winter to summer. The results are shown in the left panel of Fig. 4. The period with statistically different composites is grey shaded. Thus, the NH anticyclone in La Niña years is significantly stronger than in El Niño years within the first 5 months of the year, i.e. until MJJ. This statistical analysis indicates that the influence of ENSO on the anticyclone propagates from winter until early summer. The mean SF difference between La Niña and El Niño composites from winter to early summer is $\sim 6 \times 10^6 \, \mathrm{m^2/s}$.

15   **3.2   Walker circulation**

Complementary to SF, the divergent part of the horizontal flow can be described by the velocity potential VP and is shown in Fig. 5. Note that VP is a factor of 10 smaller than SF, which is consistent with the fact that the non-divergent rather than

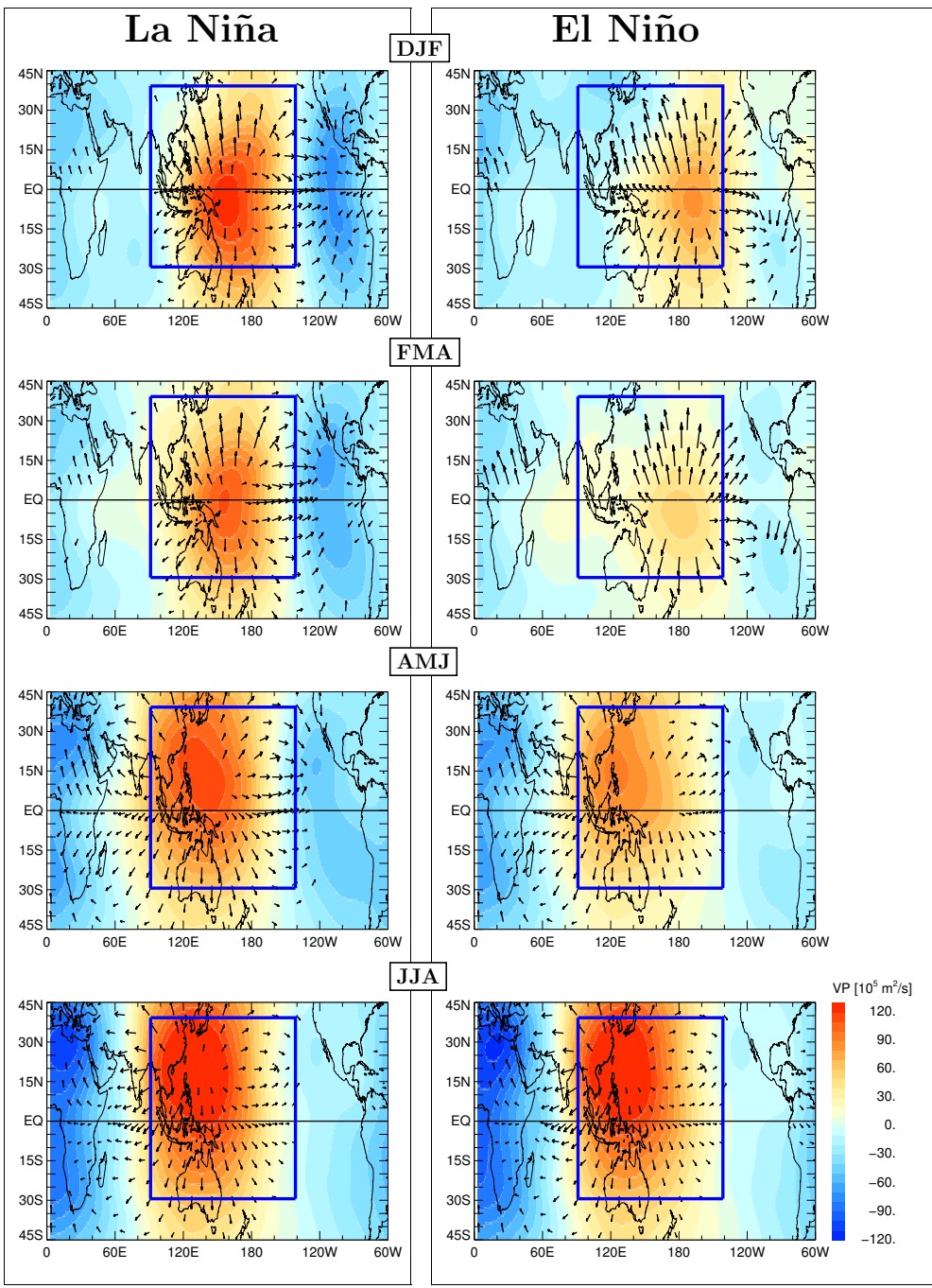

**Figure 5.** Same as Fig. 2 but for the velocity potential VP (in $10^5$ m$^2$/s) at $\theta$=380 K with arrows denoting the divergent part of the horizontal wind.

divergent part dominates the flow at $\theta$=380 K. Following Tanaka et al. (2004), the positive peak of VP indicates the intensity of the Walker circulation and the zonal mean of VP ($\overline{VP}$) quantifies the Hadley circulation (see below). The positive values of VP represent divergence or, using the continuity equation, the strength of upwelling, while the negative values are related to convergence or downwelling. In this way, the upper branch of the Walker circulation can be diagnosed in Fig. 5. The intensities

of the Walker circulation are similar to the results from Tanaka et al. (2004).

The positive peak values of VP lie in the western and central tropical Pacific for La Niña and El Niño DJF climatologies, respectively. They correspond to the locations of rising motion. The mean upwelling (downwelling) activity in La Niña winters is much stronger than in El Niño winters, in agreement with the well-known weakening of the Walker circulation after El Niño events (Wang et al., 2002). In spring (FMA) the differences between the two composites become smaller than in winter. At the

beginning of summer (AMJ), the centers of the divergence start to shift from the tropics to the extratropics and the differences become even smaller. In JJA, these centers reach the China Sea. The strength and position of the convergence/divergence centers in the La Niña composite are comparable to those of El Niño in that season.

As was done for SF, the statistical significance of the ENSO anomalies in the VP composites is diagnosed in the right panel of Fig. 4. The blue rectangle in Fig. 5, defined as [30° S, 40° N; 90° E, 140° W], represents the region of the ascending branch

of the Walker circulation. The mean positive values over this blue rectangle are calculated. The domain allows quantification of the average upwelling of the Walker circulation. The divergence in the La Niña composite is significantly higher than in the El Niño composite within the first 5 months of the year. The mean VP difference between La Niña and El Niño composites from winter to early summer is $\sim 22 \times 10^5 \, \mathrm{m^2/s}$.

### 3.3 Hadley circulation

The zonal mean of VP ($\overline{VP}$) is used to represent the Hadley circulation (Fig. 6 (a) and (b)). Note that the peak values of VP are more than three times larger than $\overline{VP}$. In winter, $\overline{VP}$ is positive in SH and negative in NH. The positive peaks represent the locations of rising air and correspond to the ITCZ. The negative peaks represent the locations of sinking air. The rising and sinking motions form the mean meridional Hadley circulation. This circulation is weaker after La Niña than after El Niño episodes, and the differences between the La Niña and El Niño composites decrease in summer.

The latitudes of positive peaks show that the rising motion is shifted southwards after El Niño winters compared to La Niña winters. Correspondingly, the ITCZ is located around 4° S and 6° S for the La Niña and El Niño composites, respectively. Fig. 6 (c) shows the difference between La Niña and El Niño composites. The upwelling and downwelling after El Niño is much stronger than after La Niña from DJF to MAM. The difference becomes smaller after AMJ. To check the statistical significance of such differences, the average rising intensity of the Hadley circulation, which is located in the tropics (from

20° S to 20° N), is calculated (Fig. 6 (d)). The values after El Niño winters are higher than after La Niña winters, especially from DJF to MAM as noted before. The mean difference is about $2\times10^5 \, \mathrm{m^2/s}$, and becomes insignificant starting from April.

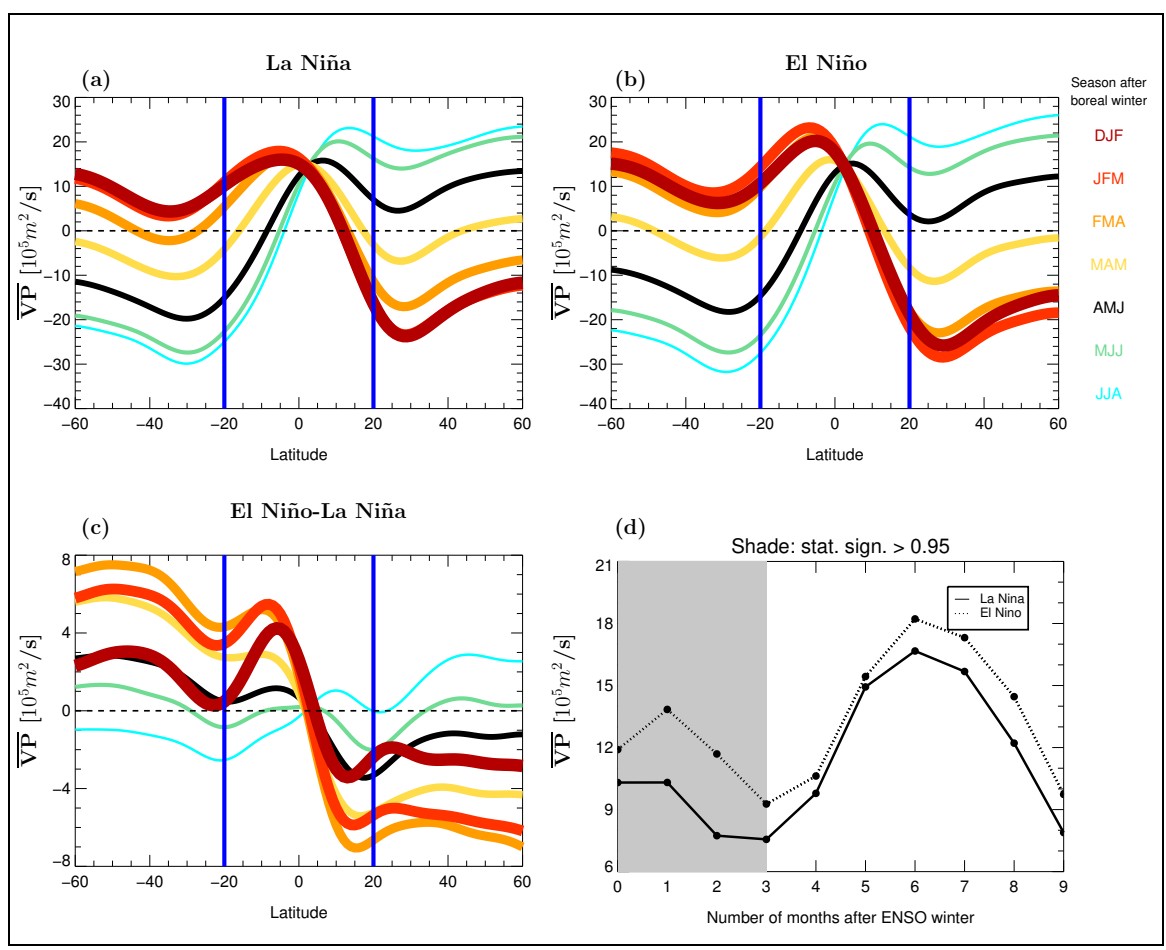

**Figure 6.** Zonal mean of the velocity potential at $\theta$=360 K defining the Hadley circulation and calculated for La Niña (a) and El Niño composites (b). The difference between La Niña and El Niño composites (c). The average intensity of Hadley circulation calculated for the domain of [20° S, 20° N] (d).

## 4 Impacts on ozone distribution

So far we have investigated the influence of ENSO anomalies on the atmospheric circulation, especially on the mean horizontal flow quantified in terms of the stream function SF (Fig. 2) and velocity potential VP (Fig. 5). Such changes of the atmospheric circulation will also affect the distribution of atmospheric constituents (Randel et al., 2009; Ziemke et al., 2015). Ozone is a sensitive indicator of transport properties in the UTLS region due to its strong vertical and horizontal gradients and its relatively long chemical lifetime. Furthermore, in the sub- and extratropics around the subtropical jet, the ozone distribution is mainly determined by transport rather than by chemistry. In this section, we quantify the impact of ENSO anomalies on the mean ozone distribution based on MLS satellite data, CLaMS simulations and SHADOZ ozonesonde data.

Particularly, we investigate now the influence of ENSO on the isentropic in-mixing of high stratospheric ozone values into the tropical TTL (Konopka et al., 2010). In the following, the ozone isoline at the tropopause is used to quantify the effect of isentropic in-mixing at $\theta = 380\,\text{K}$. Thouret et al. (2006) estimated the monthly mean climatological ozone concentration at the tropopause based on MOZAIC measurements. They found a maximum value in May (120 ppbv) and a minimum value in November (65 ppbv). Here, the isoline of 120 ppbv is used as the ozone boundary for CLaMS composites to obtain a conservative estimate of stratospheric influence. MLS ozone is high biased by $\sim$40% at 100 hPa in the tropics (Livesey et al., 2017) and even by as much as $\sim$70% inside the ASM anticyclone (Yan et al., 2016). Therefore, the isoline of 185 ppbv is used as a proxy for the tropopause in the MLS composites.

### 4.1 MLS composites

Figure 7 shows MLS ozone mixing ratio distributions at $\theta$=380 K from winter to summer after La Niña and El Niño winters. The ozone isoline at the tropopause is represented by the black solid line. During DJF and FMA, the El Niño composite is more zonally symmetric compared to La Niña. This is consistent with the less disturbed subtropical jets after El Niño winters as discussed in the last section. The region of enhanced in-mixing can be recognized as a tongue of high ozone which emerges around 120° W, 30° N during DJF and is shifted in the following months to the west until the ASM anticyclone forms.

During AMJ, this feature of in-mixing is much more pronounced for the La Niña than for the El Niño composite. This may be related to the differences in the developing process of the ASM anticyclone between La Niña and El Niño shown in Fig. 2. The mean anticyclone in AMJ is at the very beginning phase after El Niño, while ASM anticyclone after La Niña develops more quickly and the ozone distribution is affected by stronger ASM anticyclone during this period. The largest pattern difference between La Niña and El Niño ozone composites occurs during this period, while the SF shows the largest pattern difference in winter (Fig. 2, top). Ozone in-mixing anomalies seem to be delayed compared to the distribution of SF.

The black dots in Fig. 7 provide the information about regions with statistically significant differences between La Niña and El Niño composites. We can see that the differences exist almost everywhere, especially in the regions of strong in-mixing described above. During the mature phase of the ASM anticyclone (JJA), the number of black dots decreases strongly, but there is still a region of significant in-mixing differences on the ozone tongue as well as the extratropical side of the tropopause. We

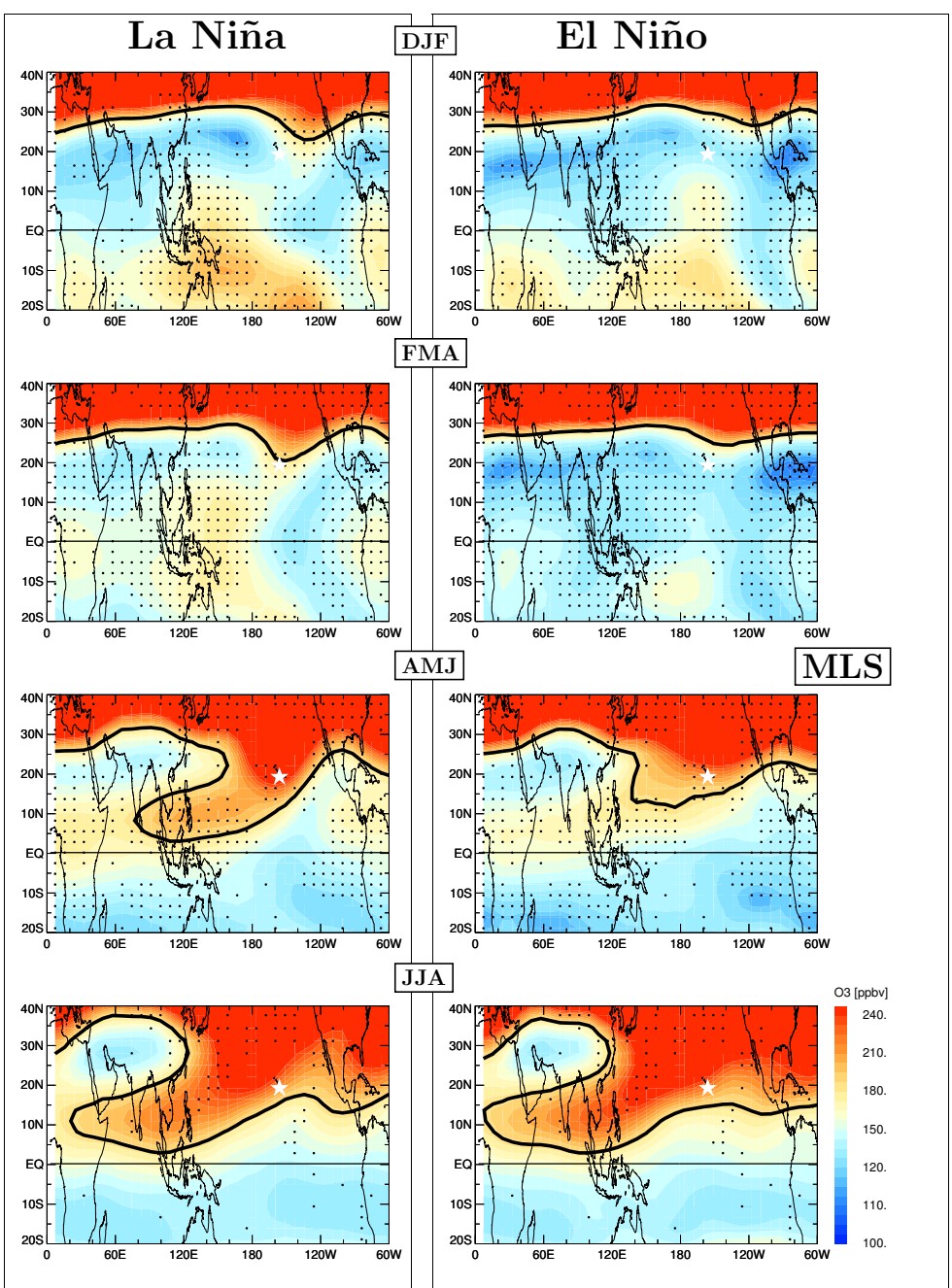

**Figure 7.** Seasonal ozone climatology derived from MLS observations (2004-2015, version 4.2) at $\theta$=380 K for La Niña and El Niño composites from winter to summer months (from top to bottom). Regions with statistically significant differences are marked by the black dots. The black isolines represent ozone of 185 ppbv, which mark the tropopause (see text).

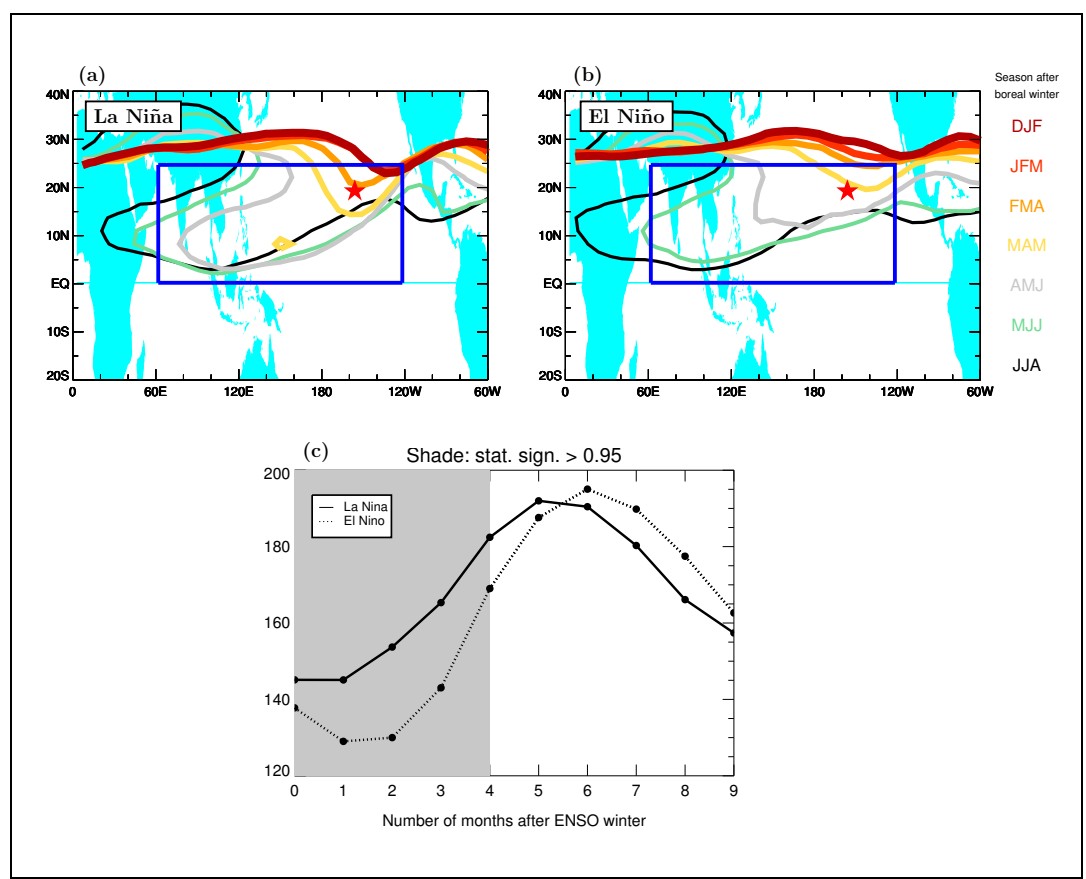

**Figure 8.** Top: Isolines of MLS ozone (185 ppbv, black lines in Fig. 7) approximating the tropopause at $\theta$=380 K for different seasons following La Niña (a) and El Niño (b) winters (from DJF (red) to JJA (black)). Bottom: The mean concentration of ozone from the blue domain in the top panel ([$0°$ N, $25°$ N; $60°$ E, $120°$ W]) marking the region of strongest ENSO-related differences in in-mixing (c).

will return to this point later. Ozone values in the center of the ASM anticyclone are lower after La Niña than after El Niño in JJA which is consistent with the similar differences in the SF (c.f. Fig. 2).

The isolines of ozone representing the tropopause are combined together in Fig. 8 (a) and (b) to illustrate the pattern of the seasonality of the ENSO-related differences in in-mixing. To quantify such differences, the mean concentration inside the
5    blue domain [$0°$ N, $25°$ N; $60°$ E, $120°$ W] is calculated and shown in Fig. 8 (c). The grey shading highlight the seasons with statistically significant differences between La Niña and El Niño composites, which are from DJF to AMJ. The average results inside the in-mixed region attest that ozone concentration after El Niño is about 16 ppbv lower than after La Niña from winter (DJF) to early summer (AMJ). The difference is a manifestation of the influence of stronger Hadley/BD circulation and weaker in-mixing after El Niño than after La Niña on the horizontal distribution of ozone around the tropopause (Randel et al., 2009;
10    Calvo et al., 2010; Konopka et al., 2016). Starting from summer, the difference of ozone distribution between El Niño and La

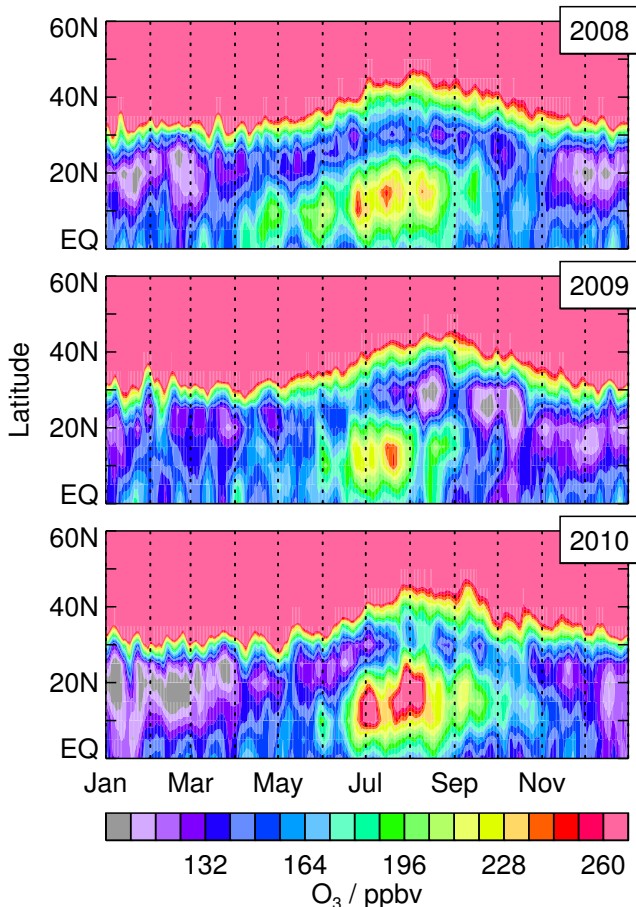

**Figure 9.** Zonally averaged ($10° - 130°$ E) time series of MLS ozone at $\theta = 380$ K (version 4.2, for more details see Santee et al. (2017)) over the course of these 3 representative years (top to bottom): 2008 (after La Niña winter), 2009 (a normal year) and 2010 (after El Niño winter).

Niña becomes statistically insignificant. Starting in JJA, the concentration of in-mixed ozone after El Niño years is even higher than after La Niña years.

To understand better such statistical differences, we investigate now in more detail the MLS observations for three example years which are representative of "typical" El Niño, La Niña and neutral conditions. Following the method described in Santee et al. (2017), we plot in Fig. 9 the time series of the zonally averaged ozone ($10° - 130°$E) at 380 K during 2008 (i.e. after La Niña), during 2009 (i.e. during a neutral year) and during 2010 (i.e. after El Niño). Over the course of these three representative years, the differences in ozone between the equator and $\sim30°$ N mainly result from different intensities of in-mixing and of the BD-circulation. Specifically, the ozone mixing ratios after El Niño winter (2010) are much lower than after La Niña winter (2008) or even during a normal year (2009), with a negative anomaly persisting from January to June,

supporting our statistical results in Fig. 7 and Fig. 8. The isentropic intrusions transport less ozone from high latitudes to the tropics following El Niño winters.

However, there is more in-mixed ozone in 2010 than in 2008 and 2009 from June to September. This could be a consequence of the differences in the BD-circulation (stronger after El Niño than after La Niña winters), which may cause higher ozone values in the northern extratropics and, consequently, stronger isentropic gradients of ozone after El Niño winters. It means that under El Niño conditions, transport of ozone-rich air from the extratropics to the tropics is inhibited during winter and spring by the strong subtropical jet, but such transport to the tropics may occur later in summer when the subtropical jet is weaker. We will come back to this point in section 5.

### 4.2 In-mixing from CLaMS

As discussed in Konopka et al. (2016, Figure 5), CLaMS reproduces fairly well the ENSO anomalies in ozone observed by MLS. However, at the time of writing the MLS composites cover only 11 years with very few strong El Niño/La Niña events. Using CLaMS ozone, we are able to extend our period to 37 years from 1979 to 2015 and obtain statistically more robust results.

Figure 10 (top) shows the same type of distribution as Fig. 8 (top) but for 37-years of CLaMS ozone simulations and with the tropopause defined by the ozone isoline with 120 ppbv. The ozone concentrations from CLaMS simulations are about 50 ppbv lower than MLS measurements at $\theta$=380 K, in part because of the zero ozone boundary condition at the ground, but they show similar patterns to MLS ozone. The CLaMS ozone distributions also show in-mixing activity over eastern and central Pacific in subsequent months following La Niña winters, with more zonally symmetric features during months following El Niño. The signatures of in-mixing over the tropical Pacific become much stronger after the onset of the ASM anticyclone (AMJ) for both composites and extend deeper into the tropics after La Niña than after El Niño winters. The differences disappear in JJA.

The largest difference between the ENSO composites exists around the eastern flank of the ASM anticyclone. To quantify this difference from CLaMS simulations, the mean concentrations in the blue domain are calculated (i.e. in the same way as for MLS) and are shown as solid black and red lines in Fig. 10 (c) for La Niña and El Niño composites, respectively (the results for the long-lasting El Niño years and for the subcomposites related to the different QBO phases are also shown and will be discussed in section 5). As for the MLS composites, the CLaMS results show a similar pattern with less in-mixed ozone after El Niño winters until early summer and more in-mixed ozone in the late summer and fall, although statistically significant differences can only be found until AMJ (grey shading). The ozone concentration after El Niño is about 12 ppbv lower than after La Niña. This difference obtained from CLaMS simulations for the time period 1979-2015 is slightly smaller than from MLS measurements for the time period 2004-2015.

### 4.3 In-mixing from SHADOZ

MLS measurements and CLaMS simulations as described above provide the ENSO-related differences in the horizontal distribution of ozone. The vertical influence of ENSO anomalies on the ozone distribution near the tropopause can also be inferred from the ozonesonde data obtained at the SHADOZ station in Hilo, Hawaii [19.43° N, 155.04° W] (marked with a star in Fig. 2,

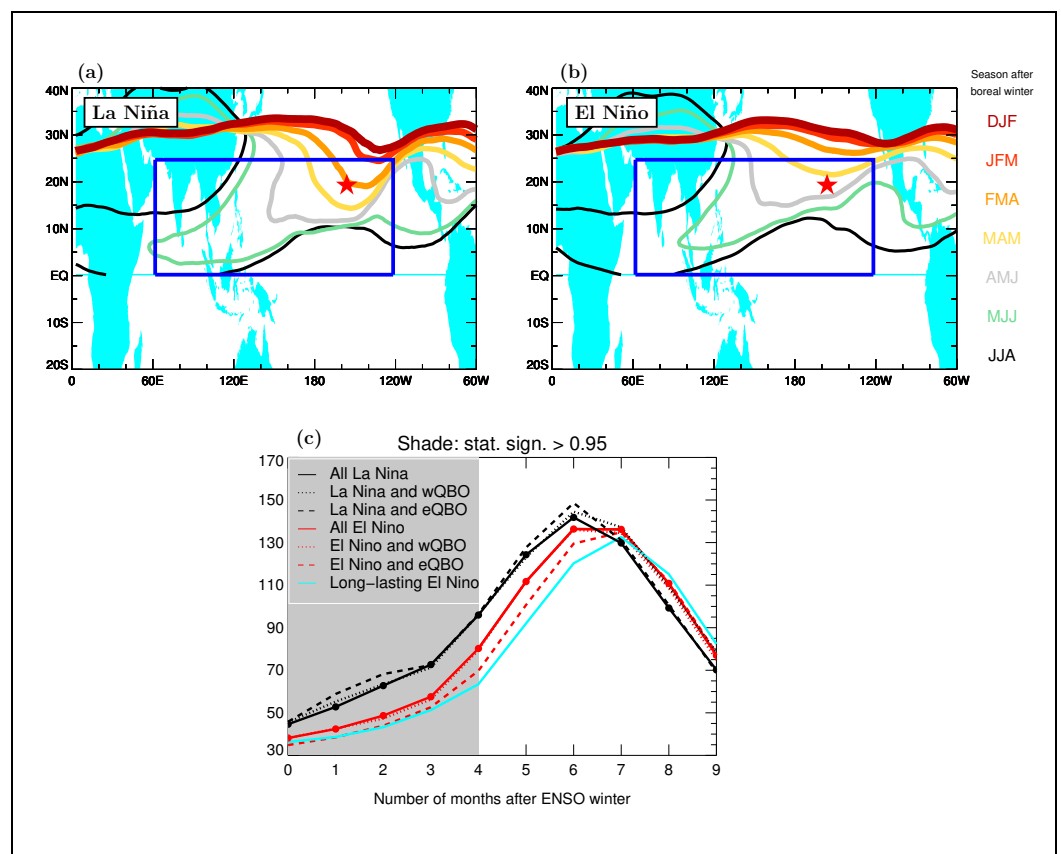

**Figure 10.** Top: Same as Fig. 8 but for CLaMS ozone with the isoline value of 120 ppbv. Bottom: Same as Fig. 8 (c) but including also the results for ENSO subcomposites with QBO westerly phase (dotted line), QBO easterly phase (dashed line) and long-lasting El Niño events (cyan line).

Fig. 7, Fig. 8, and Fig. 10) from 1998 to 2015. Hilo is located in the central Pacific at the edge of the climatological position of the anticyclone in winter (see Fig. 2). The air over Hilo is strongly affected by the meridional disruption of the subtropical jet from winter (DJF) to early summer (AMJ) following La Niña winters, while it is within the tropics following El Niño winters.

The resolution of the SHADOZ ozone profiles is not the same for the whole period, so the data is degraded to the vertical
5   resolution of 200 m for all the years to calculate the ENSO composites introduced in section 2. Figure 11 shows the ENSO-related seasonal variation of ozone with altitude over Hilo (red and black solid profiles), as well as their variability due to the QBO phase (dotted and dashed lines), which will be discussed in the next section.

The mean ozone profiles during and after El Niño show a characteristic "S"-shape structure for all the seasons, with the lowest value near the surface, a maximum near 6 km, a minimum near 12-13 km, and a subsequent increase toward stratospheric
10  values. The minimum ozone concentrations at $\sim$ 12-13 km are located at the level of main convective outflow and are therefore caused by uplift of tropospheric air (Folkins et al., 2002; Thompson et al., 2012). On the other hand, the ozone profiles from

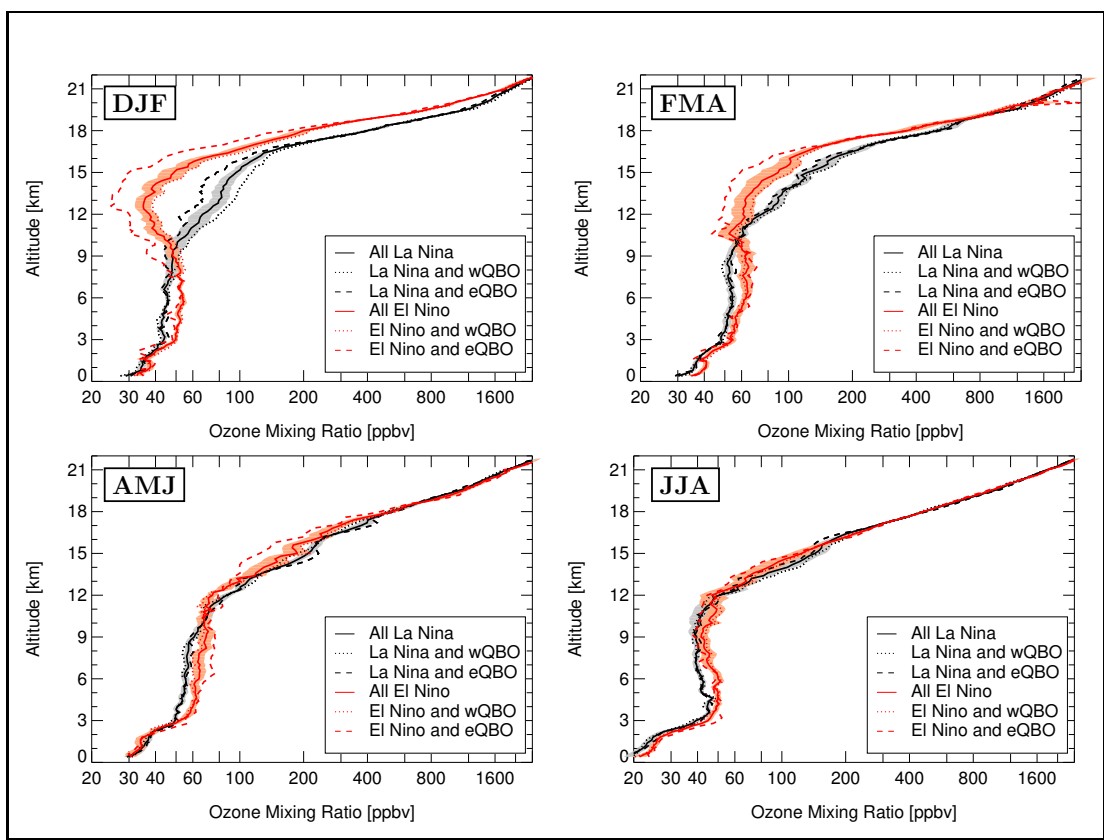

**Figure 11.** Composites of the ozonesonde measurements from SHADOZ in Hilo, Hawaii [19.43° N, 155.04° W] during 1998-2015. Black and red lines represent the seasonal mean profiles for La Niña and El Niño composites, respectively. The shading indicates the standard deviation of the mean. The dotted and dashed lines represent the results for subcomposites defined by the westerly and easterly phase of the QBO, respectively.

La Niña winters do not show such a minimum. On average, the ozone concentration for La Niña is about 44 ppbv higher than for El Niño from 9 km to 18 km in DJF (top left). The ozone concentration differences between La Niña and El Niño during FMA and AMJ (top right and bottom left) are smaller, with mean values around 38 and 20 ppbv, respectively. Finally, there is no clear difference between these two composites during JJA (bottom right).

5     The results from SHADOZ indicate that the air masses are more affected by in-mixing following La Niña years, and that this effect is not only confined to the region around 380 K ($\approx$ 15 km) but can be diagnosed throughout the whole UTLS region. Especially in winter, ENSO related anomalies in the ozone profile are quite large (from 9 km to 21 km) comparing to other seasons. The influence lasts from winter (DJF) to early summer (AMJ), but vanishes during JJA. Interestingly, the ENSO anomaly of in-mixing changes sign in the middle troposphere below 9 km. We discuss this point in the following section.

## 5 Discussion

The ENSO anomaly induces two types of variability in the global ozone distribution: On the one hand, the stronger Hadley/BD-circulation during and after El Niño winters transports less ozone into the TTL and more ozone in the extratropical lower stratosphere and, consequently, stronger latitudinal gradients of ozone on all isentropes in the UTLS region have to be expected (Randel et al., 2009; Calvo et al., 2010; Konopka et al., 2016). On the other hand, a less disturbed subtropical jet after El Niño suppresses more effectively the isentropic in-mixing of ozone into the tropics during winter and spring (this effect was extensively shown in this paper), while during late summer and fall higher ozone values, although less frequently, can be in-mixed into the TTL.

The latter effect can be seen in the MLS observations at $\theta = 380$ K (Figure 9) mainly caused by isentropic in-mixing around the eastern flank of the ASM anticyclone. This effect can also be inferred from our statistical analysis of the enhanced mean ozone values in the blue region discussed in Figure 8. These values shown in Figure 8 and 10 for MLS and CLaMS, respectively, suggest that during late summer and fall the in-mixed ozone is higher after El Niño than after La Niña winters, although we cannot prove the statistical robustness of this result. In addition, all the SHADOZ mean profiles around 3-9 km (Figure 11) show higher ozone for El Niño than for La Niña composites.

To discuss this point in more detail, Figure 12 shows from top to bottom the seasonal results of the zonal mean (120° E - 120° W) ozone anomalies after La Niña and El Niño from the surface to 70 hPa as derived from the respective CLaMS composites. The depicted ozone anomalies are mainly due to changes in the Hadley/BD-circulation, with the largest negative (positive) changes in the TTL and positive (negative) changes in the lower extratropical stratosphere mainly in the NH following El Niño (La Niña) winters. Although the largest ozone anomalies can be found in DJF and FMA, their absolute values weaken in the following months, especially in the tropics. In addition, the positive anomaly in the north of the subtropical jets (black lines in Fig. 12) under El Niño conditions propagates downwards into the middle troposphere, mainly in the NH.

We conclude that enhanced tropical upwelling in DJF and FMA following El Niño transports ozone poor air from the surface to the TTL. Likewise, the enhanced downwelling poleward of the subtropical jets following El Niño transports ozone rich air from the stratosphere to the sub- and extratropical middle troposphere. The higher ozone as observed by MLS at $\theta = 380$ K during late summer 2010 (Figure 9) as well as the higher ozone in the middle troposphere below 9 km in Hilo during DJF and FMA following El Niño (Figure 11) may be partially related to the isentropic transport of ozone-rich air from the stratosphere. While in the first case, the isentropic transport happens above the jet, mainly on the eastern flank of the ASM anticyclone, in the second case the isentropic pathway of transport is related to the isentropes below the jet, i.e. to the $\theta$ surfaces between 320 and 340 K (Newell et al., 1999; Thouret et al., 2001; Hayashi et al., 2008; Pan et al., 2015).

Inspired by the work of Chowdary et al. (2016) showing decreasing Indian summer monsoon rainfall after long-lasting El Niño events, episodes which last until the fall or over the whole year following the El Niño winters are now selected (i.e. the years 1982, 1987 and 1992 listed in Table 1). Here, we investigate whether their mean influence on the atmospheric circulation and on the ozone distribution, although not statistically significant, will increase the El Niño-related effects derived in the

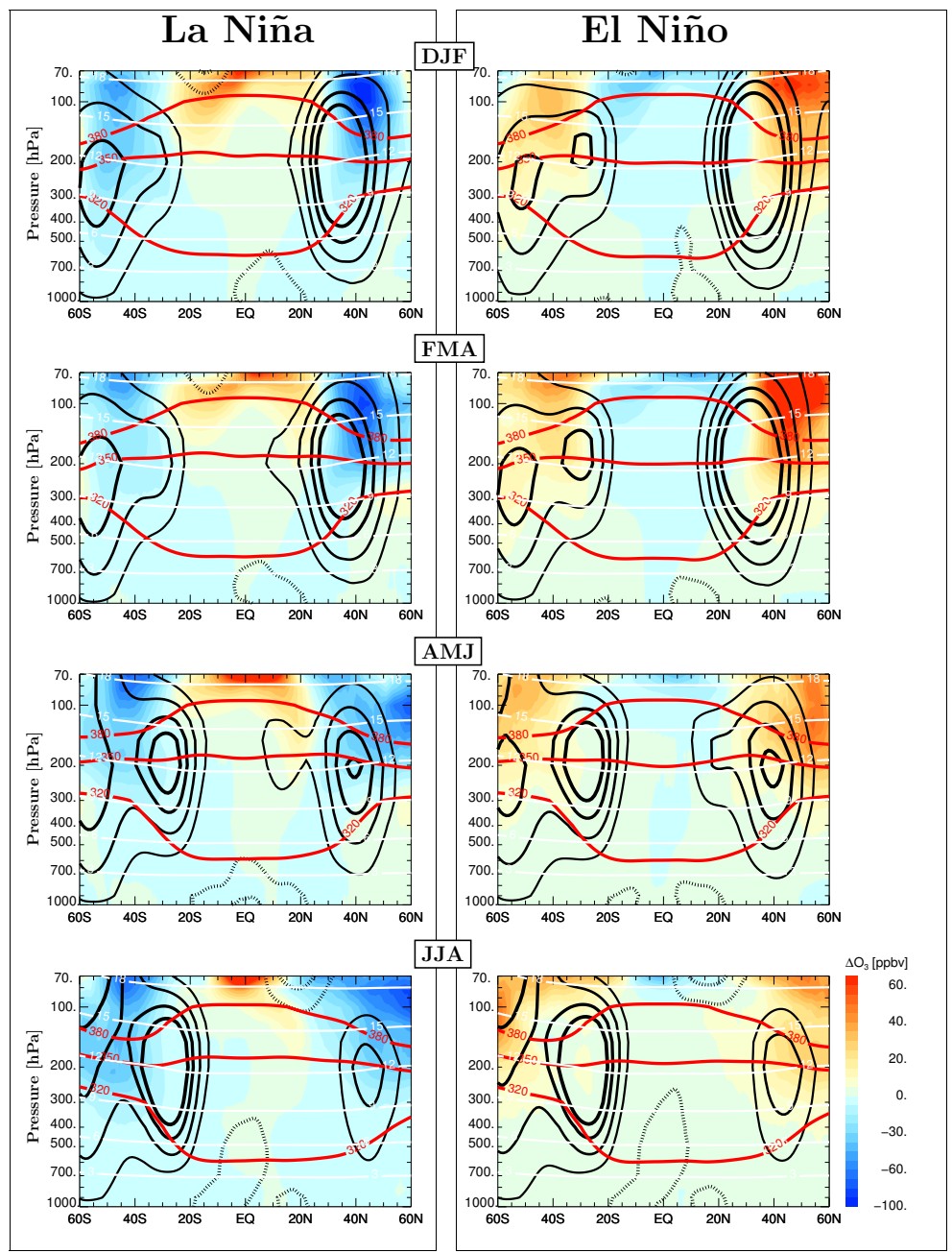

**Figure 12.** The anomalies of zonally averaged ozone in west and central Pacific [120° E, 120° W] from DJF to JJA based on CLaMS simulations covering 1979-2015. The solid and dashed lines are the zonal means of the westerlies (10, 17, 24, and 30 m/s) and easterlies (−5, −10, and −20 m/s), respectively. Red and white lines represent potential temperature (K) and geopotential height (km), respectively.

| Number of months after ENSO winter | La Niña | | | El Niño | | | long-lasting El Niño | | |
|---|---|---|---|---|---|---|---|---|---|
| | SF | VP | HC | SF | VP | HC | SF | VP | HC |
| 0 | 34 | 127 | 24 | 25 | 83 | 26 | 25 | 135 | 29 |
| 1 | 34 | 115 | 21 | 25 | 73 | 22 | 25 | 111 | 24 |
| 2 | 26 | 112 | 16 | 17 | 55 | 15 | 18 | 78 | 21 |
| 3 | 26 | 108 | 15 | 6 | 55 | 16 | 8 | 41 | 23 |
| 4 | 14 | 120 | 17 | 5 | 93 | 21 | 2 | 58 | 29 |
| 5 | 23 | 147 | 18 | 14 | 131 | 23 | 8 | 82 | 31 |
| 6 | 29 | 148 | 16 | 27 | 145 | 20 | 23 | 95 | 27 |

**Table 2.** List of the maximum strength of the NH anticyclone (SF in $10^6$ m$^2$/s), Walker circulation (VP in $10^5$ m$^2$/s) and Hadley circulation (HC in $10^5$ m$^2$/s) after La Niña, El Niño and long-lasting El Niño found inside the blue domains in Figures 2, 5 and 6, respectively.

previous sections. Table 2 shows the peak values of SF, VP and Hadley circulation found inside the blue domains in Figures 2, 5 and 6, respectively.

Indeed, SF, VP and Hadley circulation averaged over these three years show strongest anomalies if compared to all El Niño years. In particular, the ASM anticyclone is weaker and the Hadley circulation is stronger for most considered months following the long-lasting El Niño winters. The onset date of the ASM after long-lasting El Niños is even slightly later than after the other El Niño winters (Fig. 3). Accordingly, the ozone concentrations in the tropics are less disturbed by isentropic intrusions from the subtropics. Consequently, lowest ozone concentrations are detected in the blue domain in Figure 10 until the end of summer, at which time ozone following long-lasting El Niños switches to having the highest values in the early fall (cyan line in Figure 10). This indicates that if El Niño does not decay until the following summer, its influence on the ASM anticyclone and ozone will last longer.

Neu et al. (2014) found that the superposition of El Niño and easterly QBO phase increases ozone flux from the stratosphere into the troposphere, resulting in enhanced tropospheric ozone values in mid-latitudes. The opposite effect occurs for the combination of La Niña and westerly QBO phase. Motivated by this study, we investigate how the QBO phase affects our results. Table 1 shows that La Niña winters are almost equally affected by westerly and easterly QBO phases, while during El Niño the westerly QBO phase occurs more often. To quantify the potential influence of the QBO phase, we compare the difference between La Niña and El Niño subcomposites defined by the westerly and easterly phases. The CLaMS results in the blue rectangle at 380 K (Fig. 10 (c)) show that the ozone concentration after La Niña events is higher than after El Niño events during both phases of the QBO, but their difference is larger during the easterly than during the westerly QBO phase. Similarly, the SHADOZ ozone data (Figure 11) shows that the ozone concentration after La Niña events is higher (lower) than after El Niño events in the UTLS (middle troposphere) during both phases of the QBO, while the respective subcomposites show larger differences during the easterly than during the westerly phase. This indicates that our results on the ENSO effects are robust, but the difference will be enhanced (weakened) during the easterly (westerly) phase of the QBO.

# 6 Conclusions

ENSO typically shows the strongest signal in boreal winter, but it can affect the atmospheric circulation and constituent distributions until the next fall. To quantify the influence of ENSO on the atmosphere from a dynamical perspective, the stream function (SF) and the velocity potential (VP) are introduced. SF and VP represent the divergence-free and the rotation-free part of the horizontal wind field, respectively. The results show that the subtropical jets after El Niño winters are more zonally symmetric than after La Niña winters. Furthermore, the meridional disruption of the subtropical jets during El Niño are weaker compared to La Niña winters. The anticyclonic circulation in the tropics following El Niño is weaker than following La Niña. The strength of the ASM anticyclone after El Niño is slightly weaker than after La Niña in early boreal summer, and the onset date in El Niño years is about half a month later than in La Niña years. VP after El Niño is weaker than after La Niña from winter until early summer because of the weaker Walker circulation in El Niño years. The Hadley circulation after El Niño is much stronger than after La Niña from winter to spring.

The anomalies of the atmospheric circulation caused by ENSO also affect the distribution of atmospheric composition. MLS satellite measurements (2004-2015) and CLaMS simulations (1979-2015) are used to analyse the influence of ENSO on the ozone distribution in the vicinity of the tropopause (380 K). The results from CLaMS simulations show similar patterns as MLS measurements. In both, ozone patterns after La Niña winters and springs show in-mixing over the east and central Pacific, while the ozone patterns after El Niño winters and springs are more zonally symmetric. The in-mixing difference between La Niña and El Niño is striking during the onset of the ASM anticyclone (AMJ). Intrusions from the high latitude stratosphere reach much deeper into the tropics after La Niña winters than after El Niño winters. This indicates that the ozone anomaly lags the atmospheric circulation anomaly in El Niño/La Niña winters by about 4 months. Based on the ozonesonde data from SHADOZ (1998-2015) in Hilo, Hawaii, the vertical impact of ENSO on the ozone distribution is investigated. The "well known" vertical "S" shape structure only exists in the ozone profiles following El Niño but not La Niña from winter to early summer. The ozone concentration in the UTLS after El Niño is lower than after La Niña from DJF to AMJ. Our results demonstrate that the air masses over Hilo following La Niña encounter stronger (weaker) in-mixing in the UTLS (middle troposphere) compared to El Niño.

Weaker in-mixing and stronger Hadley circulation due to El Niño cause lower ozone mixing ratios in the tropical UTLS compared to La Niña from winter to early summer. However, the in-mixed ozone following El Niño winters may become higher in the subtropical middle troposphere as well as in the TTL in late summer and fall. This effect is related to a stronger Hadley/BD-circulation after El Niño compared to La Niña, which may cause higher ozone values in the extratropics and, consequently, stronger isentropic and meridional gradients of ozone after El Niño winters. The duration and intensity of the El Niño related anomalies are amplified if only the long-lasting episodes are considered. The ENSO related anomalies are enhanced (weakened) during the easterly (westerly) phase of the QBO.

*Acknowledgements.* This work was supported by the Strategic Priority Research Program of Chinese Academy of Sciences, Grant No. XDA2006010203, the National Natural Science Foundation of China, Grant No. 91337214, 41675040 and the International Postdoctoral Exchange Fellowship Program 2015 under grant No. 20151011. The European Centre for Medium-Range Weather Forecasts (ECMWF) provided meteorological analysis for this study. OLR and ENSO MEI index data is provided by NOAA. Ozonesonde data is provided through the SHADOZ database. Work at the Jet Propulsion Laboratory, California Institute of Technology, was carried out under a contract with the National Aeronautics and Space Administration. We would like to thank Suvarna Fadnavis for some discussions which motivated us to do this work. The stream function and velocity potential are calculated based on the method from H. L. Tanaka. Excellent programming support was provided by N. Thomas.

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
