# Peer review of "El Niño Southern Oscillation influence on the Asian summer monsoon anticyclone"

_Atmospheric Chemistry and Physics, 2017_

## Referee Comment (RC1) · Anonymous Referee #3 · 26 Jan 2018

General Comments

This paper examines how the Asian summer monsoon anticyclone is affected by ENSO, and what the implications are for ozone transport in the subtropical UTLS region. In general I find the paper interesting and insightful, and I appreciate the use of measurements, reanalysis, and model results to examine the role of ENSO on ozone in the ASM region. I do have some comments below to help improve the paper.

Specific comments

The colorbars have been improved for most plots since I reviewed the paper last. I think the red/blue colorbar could also be used for Figure 11, I'm not sure why that one is different.

[Figure]

Page 3, line 1-2: It's unclear to me what is meant by "We investigate how long ENSO related statistically relevant differences can be diagnosed". What is meant by long? You investigate the length of these relationships? Or is it meant to mean you look at differences over a long time period? It's not clear.

Page 3, lines 14-15. Why this particular threshold of 0.9? What is the sensitivity to threshold? (i.e., how much do the results change for different thresholds?)

Page 4, lines 3-4: Might explain more- is the significance assessed using a 2-tailed student t-test, for example?

Page 7, lines 2-5: Do you have a sense for how robust these ENSO features are relative to internal variability? The reanalysis period is still relatively short to evaluate dynamical differences. See, e.g., Deser et al. (2017).

Figure 3/4/6- indicate in caption what the different colored dots mean, if anything. Figure 3 should also have something about the dashed region being significant (it's a bit confusing because in Fig. 4 the dashed region indicates significance, however, in Figure 3 the differences should be insignificant by JJA, correct?). Finally, in Fig 3 the x-axis is labeled as "Month" but Figure 4 says "Month after DJF". Month makes more sense to me if these are indeed the months; does "Month after DJF" imply that 0=March?? Or does 0=JFM? It would help to label the axis tick marks with exactly what is meant.

Technical Edits

Page 2, line 30: "only few"->"few" or "only a few"

Page 5, Line 26: should be "mainly in the western Pacific"

Page 8, Line 3: "is by a factor" -> "is a factor"

Page 10, Line 4: "are by more than three times" ->"are more than three times"

---

## Referee Comment (RC2) · Anonymous Referee #1 · 31 Jan 2018

The authors present analyses which assess the impact of El Nino and La Nina episodes in NH winter on the atmospheric circulation and composition from the concurrent winter period to the following NH summer period. The paper is overall well written and well structured and the figures are well-composed. The impact of ENSO on the dynamcis and chemical composition of the atmosphere is interesting and suitable for publication in ACP, however to me the impact of the results is not carved out, yet. I recommend the manuscript to be accepted if the following points are considered in a revised version of the manuscript.

General comments:

In my opinion, the current "Conclusion" section is rather a summary of the previous results. I think the paper would substantially benefit if you would also state the actual

consequences of the presented results.

Please show some of the indicated/discussed differences in an extra panel/plot in the figures, for which the differences are relevant, e.g. Fig. 2. This would make the discussion and understanding of your line of arguments way easier.

In many of the analyses the differences of El Nino/La Nina decay after a couple of months and sometimes the results for El Nino seem to lag the results with respect to La Nina (e.g. Fig. 8 bottom). Please comment on how much of the observed differences is explicable through a time lag as e.g. diagnosed in the onset of the ASM anticyclone.

P.13 L.2-4: To me the differences in O3 are small in JJA in Fig.7 and the significance decreases strongly from AMJ to JJA. Further I do not see any real indication that ozone is lower after La Nina in the ASM AC in JJA. Also the statement relating this result to SF and VP is unclear (to me the differences for SF and VP are small as well). Please add plots showing the differences. This would help to follow your statements. You even state yourself that the differences are small: P7 L.6-9....Please clarify and comment on this.

P.7 L.6-9: You state that the forcing of the summer dynamics is only weakly related to the winter forcing. However, there is a well-known connection of ENSO and the Indian summer monsoon. You also include a reference (Chowdary et al 2016)regarding this issue. Further, have you checked how the summertime results change if the composites are not made using ENSO prior to the monsoon season but ENSO subsequent to the monsoon, i.e. assessing the influence of developing El Nino/La Nina events on the ASM? Assessing these differences could really improve the understanding of the connection between ENSO and the ISM. Also the ENSO phase during NH summer might influence your results, i.e. your results might depend on if the ENSO phase is transitioning from El Nino to La Nina or vice versa or if you have a no-decay El Nino (this example is even mentioned in the discussion) or no-decay La Nina.

P.18 L.24: "The change..." This paragraph is difficult to follow. Please rephrase to
make your analysis easier to follow. Also I do not understand why you are excluding in particular the ENSO events 1997/1998 and 2006/2007. Is this simply made to have a remaining set which contains an equal amount of El Nino years in QBO east and west phase? If so, I do not see why you would exclude these two events and not some others and I do not come to the conclusion that the results are robust because of this analysis. A better way might be to split the El Nino/La Nina events completely between QBO east and west and than make a comparison. (I know that this leads to a very limited number of cases, especially for the La Nina events. But you do this anyhow when you look at the no-decay El Ninos). The number of considered ENSO events could be increased by including/checking, JRA-55, which extends back to 1958.

P.11 L.2: You assess the significance of the changes of the Hadley circulation using a mean of VP from 20S to 20N. As a sensitivity: How does this change if you use a region centered around latitude of the peak value of  $\overbar{VP}$ , or a mean over a region where  $\overbar{VP}$  falls below a certain threshold?

Minor comments:

P.2 L.31-33: This sentence is strange; the part "... like weather patterns or precipitation..." seems disconnected from the previous part of the sentence.

P.3 Fig. 1: You might want to add a reference to Konopka et al. 2016; same for Table 1.

P.3 L.3: Add that you are using also O3 data from a Lagrangian CTM.

P.4 L.4: Please add a reference for the Monte Carlo method you are using and add a short explanation what the Monte Carlo method does.

P.4 L.6: You are using data for 2016 also as December 2015 is listed as El Nino month....so for the assessment of changes related to ENSO you are also using MLS, ERA-I and CLaMS data for 2016, right? Please update this in multiple occasion throughout the paper.

**ACPD**
P.5 L.4-6: Add the information of how many El Nino (La Nina) events are considered to the information of how many months contribute to the composites. IS the information on how many months are considered even important, as you show DJF composites anyhow? Do you average the months or the years when averaging the data over multiple years? Please clarify!

P.5 L.4: Please stick with one way: "6/8 ... La Nina/El Nino" or "6 (8) ... La Nina (El Nino)" throughout the whole paper.

P.5 L.9: Correct the URL for OLR data from NOAA and add statement to the acknowledgment! Probably also for the ENSO data!

P.5 L.24: "The climatological..." add "with respect to La Nina and El Nino conditions" or something like that and finish with "building the analogue composites for OLR as for the SF" or something similar.

P.5 L.26: Should this better state: "anticyclones are mainly located over the maritime continent during La Nina ... partially shifted towards the western Pacific during El Nino events.". Please clarify this sentence.

P.7 Fig.3: What does the hatching indicate in this figure? Remove if unnecessary.

P.7 L.10: Please add "climatological" after "mean" and add that the climatological mean is not shown. Otherwise it is hard to understand what you are referring to.

P.8 Fig. 4: Consider to make the hatching a little darker (for printed versions of the paper). Also in similar plots (e.g. Fig. 6).

P.8 L.17: Change to "The domain is supposed to...". How do the results change if you would check for the size of the area which lies within a specific VP contour, instead of using a fixed rectangle, where you might end up averaging positive and negative values of VP?

P.9 Fig.5: You may want to add a description for the blue rectangles and keep them
in all panels, to guide the eye. P.10 Fig.6: The different colours of the seasons are difficult to read, especially JJA MJJ and MAM. Also consider to add a plot showing the differences.

P.11 L.32: "Ozone in mixing...." I do not see why they are delayed. Please refer to the figures showing this time delay.

P.12 Fig.7: Do you show ozone isolines at 380K or at the tropopause altitude for the black contours? This is also confusing in other figure captions and multiple occasions in the text (e.g. P.13 L.6, P.14 L.9, ...). If it is at 380K simply state "at 380K". Please clarify!

P.14 L.2: I do not think that "Thus" is the right wording here.

P.15 L.10: How do CLaMS results and MLS measurements compare if you restrict the CLaMS data to the period 2005-2015?

P.17 Fig.11: I guess you are showing longitudinal averages in the west and central Pacific region of zonal anomalies. Please try to be clear (also in several parts of the text). Also add "the" in front of "..../theta = 380K...".

P.18 L.13: Change "three years" to either "two years" (1987 and 1992) or "two ENSO events" or something similar.

P.18.L14: "In particular...". Please add the time periods when the ASM (Hadley circulation) is weaker (stronger). This will also help to follow your conclusion in the following sentences, which is not clear at the moment.

Additional remarks:

P.2 L.18: Add the full version of STE.

P.3 L.8: Add full version of abbreviation NOAA = National ....

P.5 L.1: Maybe change to: "Ozone distributions are used to validate our diagnostics of
the flow and to understand the effect....in the UTLS region."

P.5 L.16: Change to: "The panels in Fig. 2 start from ..."

P.5. L.3: Remove ")(" and add "," after "(SHADOZ...".

P.5 L.30: add "during NH summer" or "during JJA" after "(ASM) anticyclone"

P.5. L.32: I guess that "asymmetric" should be changed to "antisymmetric".

P.7 L.6 : "stronger localized" do you mean "stronger and more localized" or simply "more localized"?

P.8 L.12: I guess you want to state: "In spring (FMA)" the differences between the two composites are smaller than in winter."

P.9 L.9: Add blank after "variability"

P.10 L.4: Please add (\overbar{VP}), to indicate that this is the zonal mean of VP.

P.10 L.8: Add "winters"/"events" or "episodes" after El Nino and change "with decreasing ENSO differences" to "and the differences between the El Nino and La Nina composites decrease from DJF to JJA"

P.14 L.8: I guess on should used "as" instead of "like"; this also occurs at multiple occasions in the text (e.g. P.5 L.8). Also add "(top)" after "Fig. 8".

P.15 L.21: If the sentence starts with Figure you should write "Figure" instead of "Fig." (this is easier to read and as I know this is Copernicus standard, check for other parts in the text)

P.15 L.12: Change "... simulations above ..." to "... simulations as described above ..."

---

## Author Comment (AC1) · 8 May 2018

Many thanks to Anonymous Referee 3 for thoughtful comments and suggestions that have helped to improve the presentation and completeness in this manuscript. The comments from the referee are denoted by italic letters. Our responses and a brief summary of related changes to the manuscript are given below. The references to the manuscript, in particular substantial changes in the manuscript are highlighted in red.

1. ***The colorbars have been improved for most plots since I reviewed the paper last. I think the red/blue colorbar could also be used for Figure 11, I'm not sure why that one is different.***

   A. The colorbar is changed for Figure 12 (Figure 11 in the ACPD version).

2. ***Page 3, line 1-2: It's unclear to me what is meant by "We investigate how long ENSO related statistically relevant differences can be diagnosed". What is meant by long? You investigate the length of these relationships? Or is it meant to mean you look at differences over a long time period? It's not clear.***

   A. Typically, ENSO occurs in winter, but the influence of ENSO does not just exist in winter. We investigate the time period of ENSO influence. In the text (P.5 L.1), it's changed as " we investigate how the ENSO winter signal propagates into the following seasons".

3. ***Page 3, lines 14-15. Why this particular threshold of 0.9? What is the sensitivity to threshold? (i.e., how much do the results change for different thresholds?)***

   A. Usually, the threshhold should be 1. We use the threshold of 0.9 to get larger sample. The results don't change too much if we change it as 1 (see also Konopka et al., 2016). Below we show the differences in SF using the threshold with 1. Figure RL 1 and Figure RL 2 show the strength of the anticyclone might be slightly different with the results in the manuscript if we take 1 as the threshold. However, the meridional disruption during El Niño are weaker compared to La Niña winters, the anticyclone after La Niña events is stronger than after El Niño, and the differences between La Niña and El Niño composites persist from winter to early summer. These are the same conclusions as in the draft, so our results in the manuscript are robust.

4. ***Page 4, lines 3-4: Might explain more is the significance assessed using a 2-tailed student t-test, for example?***

   A. Figure RL 3 shows the statistical difference results (black dots) between El Nio and La Nia composites from the 2-tailed student t-test from MLS ozone at 380 K. The significantly different regions from the 2-tailed student t-test are less than the regions from the Monte Carlo (Figure 7 in the draft) significance test. But the results here also show significant difference in the regions of strong in-mixing. During the mature phase of the ASM anticyclone (JJA), the number of black dots decreases strongly like in the Monte Carlo significance test. So the statistical difference results from the Monte Carlo significance test are robust.

5. ***Page 7, lines 2-5: Do you have a sense for how robust these ENSO features are relative to internal variability? The reanalysis period is still relatively short to evaluate dynamical differences. See, e.g., Deser et al. (2017).***

   A. We agree that the reanalysis period is short. We don't really know how robust the ENSO features are relative to the internal variability. This needs further investigation, but is beyond the scope of this study. We plan to use CLaMS-climate coupling model in the future. So we will return to this point in the further work.

6. ***Figure 3/4/6- indicate in caption what the different colored dots mean, if anything. Figure 3 should also have something about the dashed region being significant (it's a bit confusing because in Fig. 4 the dashed region indicates significance, however, in Figure 3 the differences should be insignificant by JJA, correct?). Finally, in Fig 3 the x-axis is labeled as "Month" but Figure 4 says "Month after***

[Figure]

Figure RL 1: Climatologies (composites) of the stream function (SF, in $10^6$ m$^2$/s) at $\theta$=380 K calculated from ERA-Interim (1979-2015) for months following La Niña (left) and El Niño (right) winters until summer (from top to bottom). The blue rectangles mark the locations of strong anticyclone in NH.

[Figure]

Figure RL 2: The average value of the stream function in the domain of [0° N, 35° N; 0° E, 160° W] for La Niña (solid line) and El Niño (dotted line) composites at $\theta$=380 K. The grey shading region denotes the period with statistically significant differences between the two composites.

*DJF". Month makes more sense to me if these are indeed the months; does "Month after DJF" imply that 0=March?? Or does 0=JFM? It would help to label the axis tick marks with exactly what is meant.*

A. The colored dots in Figure 3/4/6 meant different month or season. We changed all of them to black color in revised paper because the x-axis includes the time information. The dashed lines in Figure 3 were supposed to help to recognize the onset date difference between El Nina and La Nina. Probably it's confusing, so the hatching is removed now. The climatology of U wind is calculated from monthly composite, so the x-axis label marks the month information in Figure 3. The x-axis label represents the season information in Figure 4/6 because the climatologies are calculated from the seasonal composites. To make the figures easy to understand, we changed the x-axis labels and captions in Figure 4/6.

7. *Page 2, line 30: "only few"->"few" or "only a few"*

    A. It's changed in the text (P.2 L.32).

8. *Page 5, Line 26: should be "mainly in the western Pacific"*

    A. It's changed in the text (P.7 L.5-6).

9. *Page 8, Line 3: "is by a factor" -> "is a factor"*

    A. It's changed in the text (P.8 L.17).

10. *Page 10, Line 4: "are by more than three times" ->"are more than three times"*

    A. It's changed in the text (P.10 L.21).

[Figure]

Figure RL 3: Seasonal ozone climatology derived from MLS observations (2004-2015, version 4.2) at $\theta$=380 K for La Niña and El Niño composites from winter to summer months (from top to bottom). Regions with statistically significant differences are marked by the black dots. The black isolines represent ozone with 185 ppbv, which mark the tropopause.

---

## Author Comment (AC2) · 8 May 2018

Many thanks to Anonymous Referee 1 for the contributions, and particularly for raising several pertinent questions that have helped to improve the clarity of the manuscript. The main changes include the influence of long-lasting El Niño and QBO on the results. In the following, we address all the points raised in the review (denoted by italic letters). The references to the manuscript as well as substantial changes in the manuscript are highlighted in red.

1. *In my opinion, the current "Conclusion" section is rather a summary of the previous results. I think the paper would substantially benefit if you would also state the actual consequences of the presented results.*

   A. We did substantial changes in the "Discussion" and "Conclusions" section. The "Conclusions" section is completely reorganized, it includes both the results and brief explanation about the causes of ENSO related anomalies.

2. *Please show some of the indicated/discussed differences in an extra panel/plot in the figures, for which the differences are relevant, e.g. Fig. 2. This would make the discussion and understanding of your line of arguments way easier.*

   A. In the revised draft, the results from the long-lasting El Niño years are included (see Figure 3, Figure 10, and Table 2). The main conclusion about the long-lasting El Niño events can be summarized as "The duration and intensity of El Niño related anomalies may be reinforced through the late summer and fall if the El Niño conditions last until the following winter".

   The influence of QBO phases on the results are shown in Figure 10 and Figure 11 in the manuscript. The text is also adjusted in the discussion section, correspondently (P21 L.11-22). The main result is that the ENSO related anomalies are enhanced (weakened) during the easterly (westerly) phase of the QBO. The details are included in the text.

   The figures below show some extra results from the long-lasting El Niño years. The SF (Figure RL 1) and VP (Figure RL 2) distributions show strongest anomalies during the long lasting El Niño years compared to all El Niño years. Figure RL 3 shows strongest Hadley circulation during long lasting El Niño years.

   Accordingly, the ozone concentrations in the tropics show weak intrusions from the subtropics during summer (Figure RL 4). This indicates that if El Niño does not decay until the following summer, the influence of El Niño on the ASM anticyclone and ozone will last longer.

3. *In many of the analyses the differences of El Nino/La Nina decay after a couple of months and sometimes the results for El Nino seem to lag the results with respect to La Nina (e.g. Fig. 8 bottom). Please comment on how much of the observed differences is explicable through a time lag as e.g. diagnosed in the onset of the ASM anticyclone.*

   A. The draft (Figure 3) shows that the onset date of the ASM anticyclone after El Niño is about a half month later than after La Niña. We diagnose the time lag of MLS ozone after El Niño based on the same region ([5° N, 20° N; 40° E, 120° E]) as the zonal wind in Figure RL 5. Because of the difference of in-mixing and the difference in onset date of ASM anticyclone, the ozone concentrations in this domain after El Niño winters are lower than after La Niña winters. We take the onset date calculated from zonal wind (see Figure 3 in the draft)for La Niña years, the ozone concentration following El Niño winters reach the same ozone value as following La Niña winters after $\sim 20$ days later. This can be the approximate estimate of the time lag diagnosed in the onset of the ASM anticyclone from the observation.

4. *P.13 L.2-4: To me the differences in O3 are small in JJA in Fig.7 and the significance decreases strongly from AMJ to JJA. Further I do not see any real indication that ozone is lower after La Nina in the ASM AC in JJA. Also the statement relating this result to SF and VP is unclear (to me the differences for SF and VP are small as well). Please add plots showing the differences. This would help to follow your*

[Figure]

Figure RL 1: Stream function (SF, in $10^6$ m$^2$/s) at $\theta$=380 K for months following long lasting El Niño (1987 and 1992) winter until summer (from top to bottom). The arrows represent the rotational horizontal wind. White isolines indicate the strong convection regions based on OLR (thick and thin lines represent 210 and 220 W/m$^2$ contours) data, respectively.

*statements. You even state yourself that the differences are small: P7 L.6-9....Please clarify and comment on this.*

A. You are right, the significance indeed decreases strongly. We emphasized the decrease of the significance before. We agree with your point, so we changed it as "During the mature phase of the ASM anticyclone (JJA), the number of black dots decreases strongly, but there is still a region of significant in-mixing differences on the ozone tongue as well as the extratropical side of the tropopause." in the text (P.12 L.31-32) now. The MLS ozone in JJA (Figure 7, bottom) shows that the region of ASM anticyclone after La Niña events is larger than after El Niño, and the ozone in this region after La Niña events is lower than after El Niño events. The difference between La Niña and El Niño composites is small for SF and ozone distribution in JJA. Our results show that the difference can last from DJF to MJJ, but it's not significant in JJA. We removed VP-related arguments from the manuscript because the ozone distribution is more affected by the SF patterns.

5. *P.7 L.6-9: You state that the forcing of the summer dynamics is only weakly related to the winter forcing. However, there is a well-known connection of ENSO and the Indian summer monsoon. You also include a reference (Chowdary et al 2016)regarding this issue. Further, have you checked how the summertime results change if the composites are not made using ENSO prior to the monsoon season but ENSO subsequent to the monsoon, i.e. assessing the influence of developing El Nino/La Nina events on the ASM? Assessing these differences could really improve the understanding of the connection between ENSO and the ISM. Also the ENSO phase during NH summer might influence your results, i.e. your results might depend on if the ENSO phase is transitioning from El Nino to La Nina or vice versa or if you have a no-decay El Nino (this example is even mentioned in the discussion) or no-decay La Nina.*

A. From Figure 7 (JJA, bottom), we can see that the influence from ENSO is significantly decreased from the large scale perspective, but the difference (the dots in the figure) still exists from the small scall per-spective (such as India, strong in-mixing region). We didn't investigate the results based on different ENSO

[Figure]

Figure RL 2: Same as Figure RL 1 but for the velocity potential VP (in $10^5$ m$^2$/s) at $\theta$=380 K with arrows denoting the divergent part of the horizontal wind.

[Figure]

Figure RL 3: Zonal mean of the velocity potential at $\theta$=360 K defining the Hadley circulation and calculated for long lasting El Niño years (1987 and 1992). The average intensity of Hadley circulation calculated for the domain of [20° S, 20° N] (bottom).

[Figure]

Figure RL 4: Isolines of CLaMS ozone (120 ppbv at $\theta$=380 K for different seasons following long lasting El Niño (b) winters (from DJF (red) to JJA (grey)).

phase because we don't have sufficiently long period sample as in the paper from Chowdary et al. 2016 (1876-2007). This is why we put the influence of long-lasting El Niñoevents in the discussion, not in a new section.

6. ***P.18 L.24: "The change..." This paragraph is difficult to follow. Please rephrase to make your analysis easier to follow. Also I do not understand why you are excluding in particular the ENSO events 1997/1998 and 2006/2007. Is this simply made to have a remaining set which contains an equal amount of El Nino years in QBO east and west phase? If so, I do not see why you would exclude these two events and not some others and I do not come to the conclusion that the results are robust because of this analysis. A better way might be to split the El Nino/La Nina events completely between QBO east and west and than make a comparison. (I know that this leads to a very limited number of cases, especially for the La Nina events. But you do this anyhow when you look at the no-decay El Ninos). The number of considered ENSO events could be increased by including/checking, JRA-55, which extends back to 1958.***

A. We agree with your suggested method to investigate the influence of QBO phase. Figure 10 and Figure 11 in the draft show the results during westerly and easterly phase. We also change the statement in the text (for details, see the "Discussion" section). The main change can be summarized like "The results at 380 K from CLaMS simulations show that ozone concentration after La Niña events is higher than after El Niño events during both phases of QBO, but the difference between the two composites during easterly phase is lager than during westerly phase. The SHADOZ ozone data shows that the ozone concentration after La Niña events is higher (lower) than after El Niño events in the UTLS (middle troposphere) during both phases of QBO, while the two composites during easterly phase also show larger difference comparing to westerly phase. This indicates that our results about the ENSO effects are robust, but the difference will be enhanced (weakened) during easterly (westerly) phase".

7. ***P.11 L.2: You assess the significance of the changes of the Hadley circulation using a mean of VP from 20S to 20N. As a sensitivity: How does this change if you use a region centered around latitude of the peak value of \overbar{VP}, or a mean over a region where overbarVP falls below a certain threshold?***

A. We checked the significance of the changes of the Hadley circulation using a mean of VP in the domain of [5° S, 5° N], [10° S, 10° N], [15° S, 15° N], [20° S, 20° N], [25° S, 25° N] and [30° S, 30° N]. The results show that the significant difference can last from DJF to FMA in the domain of [5° S, 5° N], [10° S, 10° N] and [15° S, 15° N], and the significant difference can last from DJF to MAM in the domain of [20° S, 20° N],

[Figure]

Figure RL 5: Mean ozone in the tropics over south-east Asia ([5° N, 20° N; 40° E, 120° E]) following La Niña and El Niño winters at 380 K. The zero mark in x-axis denotes the middle of the DJF season (i.e., 15 January).

[25° S, 25° N] and [30° S, 30° N]. We choose the domain of [20° S, 20° N] because it includes all the peak locations from DJF to JJA.

8. *P.2 L.31-33: This sentence is strange; the part "... like weather patterns or precipitation..." seems disconnected from the previous part of the sentence.*

   A. It's changed as "This is in contrast with a large number of investigations connecting ENSO with the tropospheric variability of the ASM, such as weather patterns and precipitation, which have a long tradition starting with the pioneering studies of Walker (1923) and Bjerknes (1969)." in the manuscript (P.2 L.33-35).

9. *P.3 Fig. 1: You might want to add a reference to Konopka et al. 2016; same for Table 1.*

   A. The data from Figure 1 and Table 1. is extended from the period 1979-2013 to period 1979-2015. We also include "As in Konopka et al. (2016)" in the text for Figure 1 and Table 1.

10. *P.3 L.3: Add that you are using also O3 data from a Lagrangian CTM.*

    A. "Lagrangian model ozone data" is included in the text (P.3 L.5).

11. *P.4 L.4: Please add a reference for the Monte Carlo method you are using and add a short explanation what the Monte Carlo method does.*

    A. The reference and short explanation are included in the text (P.4 L.10-12) like "Monte Carlo significance test procedures consist of the comparison of the observed data with random samples generated in accordance with the hypothesis being tested (Hope, 1968)".

12. *P.4 L.6: You are using data for 2016 also as December 2015 is listed as El Nino month....so for the assessment of changes related to ENSO you are also using MLS, ERA-I and CLaMS data for 2016, right? Please update this in multiple occasion throughout the paper.*

    A. The data from 2016 is not included in our research. We use the data until the winter of 2015.

13. ***P.5 L.4-6: Add the information of how many El Nino (La Nina) events are considered to the information of how many months contribute to the composites. IS the information on how many months are considered even important, as you show DJF composites anyhow? Do you average the months or the years when averaging the data over multiple years? Please clarify!***

A. The information of how many El Nino (La Nina) events is included in the text like "MLS measurements provide 8/6 months of data for the 3/3 La Niña/El Niño episodes from 2004 to 2015. Respectively there are 14/11 months of data for the 5/5 La Niña/El Niño events from SHADOZ ozondesondes covering the period 1998-2015". Because ENSO doesn't always show strong signal in the whole DJF, we also include the months in our draft. We average the months when averaging the data over multiple years.

14. ***P.5 L.4: Please stick with one way: "6/8 ... La Nina/El Nino" or "6 (8) ... La Nina (El Nino)" throughout the whole paper.***

A. The expression is changed throughout the paper (P.5 L13-15). It's like "MLS measurements provide 8/6 months of data for the 3/3 La Niña/El Niño episodes from 2004 to 2015. Respectively there are 14/11 months of data for the 5/5 La Niña/El Niño events from SHADOZ ozondesondes covering the period 1998-2015".

15. ***P.5 L.9: Correct the URL for OLR data from NOAA and add statement to the acknowledgment! Probably also for the ENSO data!***

A. The URL for OLR is corrected in the test with "https://www.esrl.noaa.gov/psd/data/gridded/data.interp_OLR.h (P.5 L19). The statement is included in the acknowledgment like "OLR data is provided by NOAA.".

16. ***P.5 L.24: "The climatological..." add "with respect to La Nina and El Nino conditions" or something like that and finish with "building the analogue composites for OLR as for the SF" or something similar.***

A. The expression is changed as "The climatological sources of heat can be approximated by the lowest values of the OLR. The analogous composites for OLR (magenta contours in Figure 2) as for the SF are built with respect to La Niña and El Niño conditions" in the text (P.7 L3-4).

17. ***P.5 L.26: Should this better state: "anticyclones are mainly located over the maritime continent during La Nina ... partially shifted towards the western Pacific during El Nino events.". Please clarify this sentence.***

A. Here we use the OLR data to indicate the heating source of the anticyclone. So this statement is about the different locations of the heating source during La Niña and El Niño years not the locations of anticyclone.

18. ***P.7 Fig.3: What does the hatching indicate in this figure? Remove if unnecessary.***

A. The hatching was supposed to help to know the onset date of the ASM and recognize the onset date difference between El Nina and La Nina. Probably it's confusing, so the hatching is removed now.

19. ***P.7 L.10: Please add "climatological" after "mean" and add that the climatological mean is not shown. Otherwise it is hard to understand what you are referring to.***

A. The "climatological" is added in the text (P.8 L1). Here, the mean climatological anticyclone in AMJ means the climatological after El Nino and La Nina events, which is showed in the figure.

20. ***P.8 Fig. 4: Consider to make the hatching a little darker (for printed versions of the paper). Also in similar plots (e.g. Fig. 6).***

A. The hatching is changed to grey shading for Figure 4, Figure 6, Figure 8 and Figure 10.

21. **P.8 L.17: Change to "The domain is supposed to...". How do the results change if you would check for the size of the area which lies within a specific VP contour, instead of using a fixed rectangle, where you might end up averaging positive and negative values of VP?**

A. The text (P.10 L14-16) is changed as "The blue rectangle in Fig. 5, defined as [30° S, 40° N; 90° E, 140° W], represents the region of the ascending branch of the Walker circulation. The mean positive values over this blue rectangle are calculated. The domain allows quantification of the average upwelling of the Walker circulation". Here, we just average the positive values in the domain to get the intensity of upwelling. We clarify this in the text now. The statistical results don't change if we change the size of the area.

22. **P.9 Fig.5: You may want to add a description for the blue rectangles and keep them in all panels, to guide the eye. P.10 Fig.6: The different colours of the seasons are difficult to read, especially JJA MJJ and MAM. Also consider to add a plot showing the differences.**

A. The description is included in the text (P.10 L14-16) like "The blue rectangle in Fig. 5, defined as [30° S, 40° N; 90° E, 140° W], represents the region of the ascending branch of the Walker circulation. The mean positive values over this blue rectangle are calculated. The domain allows quantification of the average upwelling of the Walker circulation". The blue rectangles are added for all panels. The colours in Figure 6 are changed, and the plot about the difference is included in Figure 6 (c).

23. **P.11 L.32: "Ozone in mixing...." I do not see why they are delayed. Please refer to the figures showing this time delay.**

A. Comparing to SF, which the largest difference occurs in DJF, the largest difference of ozone in mixing occurs in AMJ. The text is changed correspondingly (P.12 L28).

24. **P.12 Fig.7: Do you show ozone isolines at 380K or at the tropopause altitude for the black contours? This is also confusing in other figure captions and multiple occasions in the text (e.g. P.13 L.6, P.14 L.9, ...). If it is at 380K simply state "at 380K". Please clarify!**

A. The colorbar shows the ozone distribution at 380 K, but the ozone isolines represent the tropopause altitude. The description is slightly changed in the text as well.

25. **P.14 L.2: I do not think that "Thus" is the right wording here.**

A. The description is changed in the text (P.12 L30).

26. **P.15 L.10: How do CLaMS results and MLS measurements compare if you restrict the CLaMS data to the period 2005-2015?**

A. The ozone concentration after El Niño is about 11 ppbv lower than after La Niña if we restrict the CLaMS data to the period 2005-2015, which is similar to the results (12 ppbv) from the period 1979-2015.

27. **P.17 Fig.11: I guess you are showing longitudinal averages in the west and central Pacific region of zonal anomalies. Please try to be clear (also in several parts of the text). Also add "the" in front of "..../theta = 380K...".**

A. Yes, we are showing the longitudinal averages in the west and central Pacific region of zonal anomalies. See the new caption in P.20 Figure 12.

28. **P.18 L.13: Change "three years" to either "two years" (1987 and 1992) or "two ENSO events" or something similar.**

A. The long lasting El Nino year 1982 is also included, so the text is changed as "three years" (P.21 L3).

29. **P.18.L14: "In particular...". Please add the time periods when the ASM (Hadley circu- lation) is weaker (stronger). This will also help to follow your conclusion in the following sentences, which is not clear at the moment.**

A. The time period is included and results are changed in the text (see P.21 L.3-10).

30. **P.2 L.18: Add the full version of STE.**

A. The full version of STE (stratosphere-troposphere exchange) is included in the text (see P.2 L.16).

31. **P.3 L.8: Add full version of abbreviation NOAA = National ....**

A. The full version of NOAA (National Oceanic and Atmospheric Administration) is included in the text (see P.3 L.10).

32. **P.5 L.1: Maybe change to: "Ozone distributions are used to validate our diagnostics of the flow and to understand the effect....in the UTLS region."**

A. It's changed in the text (see P.5 L.10).

33. **P.5 L.16: Change to: "The panels in Fig. 2 start from ..."**

A. It's changed in the text (see P.5 L.26).

34. **P.5. L.3: Remove ")(" and add "," after "(SHADOZ...".**

A. It's changed in the text (see P.5 L.12).

35. **P.5 L.30: add "during NH summer" or "during JJA" after "(ASM) anticyclone"**

A. "during NH summer" is included after "(ASM) anticyclone" (see P.7 L.9-10).

36. **P.5. L.32: I guess that "asymmetric" should be changed to "antisymmetric".**

A. It's changed as anti-symmetric in the text (see P.7 L.11).

37. **P.7 L.6 : "stronger localized" do you mean "stronger and more localized" or simply "more localized"?**

A. It's changed as "stronger and more localized" in the text (see P.7 L.17-18).

38. **P.8 L.12: I guess you want to state: "In spring (FMA)" the differences between the two composites are smaller than in winter."**

A. Yes, the text is changed (see P.10 L.9).

39. **P.9 L.9: Add blank after "variability"**

A. The blank is added in the text (see P.3 L.11).

40. **P.10 L.4: Please add ($\overline{VP}$), to indicate that this is the zonal mean of VP.**

A. $\overline{VP}$ is added in the text (see P.10 L.20).

41. **P.10 L.8: Add "winters"/"events" or "episodes" after El Nino and change "with decreasing ENSO differences" to "and the differences between the El Nino and La Nina composites decrease from DJF to JJA"**

A. It's changed as "This circulation is weaker after La Niña than after El Niño episodes, and the differences between the La Niña and El Niño composites decrease in summer" in he text (see P.10 L.23-24).

42. ***P.14 L.8: I guess on should used "as" instead of "like"; this also occurs at multiple occasions in the text (e.g. P.5 L.8). Also add "(top)" after "Fig. 8".***

A. "like" is replaced as "as" in all the occasions in the text text (e.g. P.16 L.14).

43. ***P.15 L.21: If the sentence starts with Figure you should write "Figure" instead of "Fig." (this is easier to read and as I know this is Copernicus standard, check for other parts in the text)***

A. The "Fig." is replaced as "Figure" if it's at the beginning of the sentence in the whole paper.

44. ***P.15 L.12: Change "... simulations above ..." to "... simulations as described above ..."***

A. It's changed in the text (see P.16 L.31).